# CA3 hippocampal synaptic plasticity supports ripple physiology during memory consolidation

Hajer El Oussini[1,4], Chun-Lei Zhang[1,3,4], Urielle François [1,4], Cecilia Castelli[1], Aurélie Lampin-Saint-Amaux[1], Marilyn Lepleux[1], Pablo Molle[1], Legeolas Velez[1], Cyril Dejean[2], Frederic Lanore [1], Cyril Herry [2], Daniel Choquet [1,4] & Yann Humeau [1,4] ✉

The consolidation of recent memories depends on memory replays, also called ripples, generated within the hippocampus during slow-wave sleep, and whose inactivation leads to memory impairment. For now, the mobilisation, localisation and importance of synaptic plasticity events associated to ripples are largely unknown. To tackle this question, we used cell surface AMPAR immobilisation to block post-synaptic LTP within the hippocampal region of male mice during a spatial memory task, and show that: 1- hippocampal synaptic plasticity is engaged during consolidation, but is dispensable during encoding or retrieval. 2- Plasticity blockade during sleep results in apparent forgetting of the encoded rule. 3- In vivo ripple recordings show a strong effect of AMPAR immobilisation when a rule has been recently encoded. 4- In situ investigation suggests that plasticity at CA3-CA3 recurrent synapses supports ripple generation. We thus propose that post-synaptic AMPAR mobility at CA3 recurrent synapses is necessary for ripple-dependent rule consolidation.

The importance of activity-dependent synaptic plasticity (SP) in the process of memorization is generally admitted. This statement is supported by multiple studies reporting the behavioral impact of pharmacological treatments targeting SP-related molecular mechanisms, SP-related genetic inactivation, or more recently, molecular approaches specifically affecting long-term potentiation[1–3]. The examination of the physiological consequences of these inactivation strategies in living animals generally point to alterations in behavioral performance at testing time[2,3]. However, the impact depends on the type of memory and the extent of molecular manipulations onto one or more relevant brain regions[2]. Recent reports also challenged the link existing between neurobiological consequences of synaptic plasticity blockade and animal performances. For example, Kaganovsky et al. recently reported that blocking synaptic plasticity in the CA1 region of

the hippocampus did not impact animal performance during behavioral tests, while some cellular and network proxies of learning were indeed affected in the hippocampus[2]. This may result from functional redundancy among brain regions to achieve cognitive robustness, but it also certainly complicates the interpretation of the effects of these SP interference methods on memory process.

One intriguing possibility would be that such a process of memory that includes multiple steps—encoding, consolidation and retrieval—would correspond to sequential steps of synaptic plasticity formation and maintenance[4,5]. There, synaptic tagging would be the immediate response to coincident neuronal activities supporting rapid adaptation of animal behavior in response to new situations. Subsequently, synaptic capture, necessary for the maintenance of plasticity, would occur during quiescent—awake or sleeping—states, enabling

[1]University of Bordeaux, CNRS, IINS, UMR 5297, F-33000 Bordeaux, France. [2]University of Bordeaux, INSERM, Neurocentre Magendie, U1215, F-33000 Bordeaux, France. [3]Present address: Sorbonne Université, CNRS, INSERM, Institut de Biologie Paris Seine (IBPS), Neurosciences Paris Seine (NPS), Team Synaptic Plasticity and Neural Networks, F-75005 Paris, France. [4]These authors contributed equally: Hajer El Oussini, Chun-Lei Zhang, Urielle François, Daniel Choquet, Yann Humeau. ✉e-mail: yann.humeau@u-bordeaux.fr

consolidation of a memory that can be retrieved afterward. Along this line, the hippocampal ripples may emerge as a central factor. Ripples intrinsically possess the capacity to replay the behaviorally relevant spatial sequences encoded during the awake state[6], thereby serving as the prerequisite for synaptic capture. In addition, through the broadcasting of this information, the ripple events enable the expression of SP-related molecules critical for synaptic capture[4]. Recent findings confirmed that ripple content depends on recently acquired memories[7], reactivating neuronal ensembles in cortices, such as those implicated during the performance of specific rules[8].

Physiologically, hippocampal ripples are short network oscillations at 150–250 Hz corresponding to synchronized neuronal activation that also generates synaptic waves that can be observed if recording in the dendritic fields—i.e., *stratum radiatum*—of the hippocampal CA3 and CA1 regions[6]. Interestingly, ripples can reach cortical regions—through direct or indirect projections—where they synchronize with other sleep-related oscillations, such as spindle and delta waves[6], a process that is reinforced by newly encoded learning[9]. In the hippocampus, ripples are generated in structures—such as in hippocampal CA3 region—rich in recurrent connectivity and depend both on excitatory and inhibitory local inputs that constitute the feedback loops necessary for fast oscillations building[10]. Therefore, impairing[11] or prolonging[7] ripples can be achieved by manipulating specific interneuron populations.

Even if in situ preparations do allow spontaneous oscillations that share numerous characteristics with ripples recorded in vivo[10,12,13], the relation between synaptic plasticity and ripple physiology has not yet been explored specifically in situ by using methods avoiding confounding factors such as effects on basal glutamatergic transmission[1].

Twenty years ago, we and others uncovered that AMPAR, along with NMDA and GABAA receptors, are highly mobile at the neuronal surface, and are reversibly stabilized at synaptic site due to protein/protein interactions with various synaptic partners[14]. Recently, we showed that immobilization of GluA2-containing AMPAR leads to blockade of long-term potentiation[15] without affecting basal synaptic transmission, offering a tool to assess the role of hippocampal AMPAR mobility (AMPARM) in the various phases of recent memories. We here used two alternative strategies to abolish synaptic plasticity depending on AMPARM: the first method, based on multivalent IgGs directed against the GluA2 subunit of AMPAR, has proven to be efficient in blocking LTP at Schaeffer collaterals to CA1 synapses[15]. The second one, alternatively, used specific biotinylation of GluA2 subunit and tetravalent neutravidin in the external medium to block AMPARM. This manipulation did not affect basal transmission but led to a complete absence of LTP[16]. We thus used these two strategies in the dorsal hippocampus to explore the link between synaptic plasticity, ripple physiology and learning and memory processing.

## Results

### AMPAR immobilization in the dorsal hippocampus impairs memory consolidation

Based on our previous reports showing that AMPAR immobilization at neuronal surface blocked efficiently post-synaptic expression of hippocampal LTP, we used intra-hippocampal infusions of AMPAR cross-linkers to test for the implication of synaptic plasticity in the process of memorizing a spatial alternation task. For this, mice were cannulated bilaterally above the dorsal hippocampus, and trained for working memory-based Delayed Spatial Alternation task (DSA[17]). In this task, food-deprived mice are taught to find food rewards in a Y-maze according to a simple rule: reward location is alternating between right/left ending arms (Fig. 1a). A delay of 30 s between consecutive runs is imposed, forcing the animals to remember the previous location before engaging in the upcoming run. In control conditions, a training day of 4 sessions—about 40 trials or reward positions—is sufficient for the animals to decrease their number of errors and reach

their maximal performance, which is maintained the following days (Fig. 1d). To mediate AMPAR immobilization in the dorsal hippocampus, we performed bilateral, intracerebral injections of AMPAR cross-linkers (anti-GluA2 IgGs) or their controls (anti-GluA2 monovalent Fabs) at key times of the learning process, with a sufficient delay (>1 h) between IgG injections and behavioral testing to allow efficient AMPARM blockade (Fig. 1b, c): immediately before the first learning session of day 1 (pre-learning), immediately after the end of the first training day (pre-rest), and immediately before the first session of day 2 (pre-test). Our aim was to test the importance of hippocampal AMPARM-dependent plasticity in the encoding, the consolidation and the recall of DSA rule, respectively. Collectively, our results pointed to an impact of AMPAR cross-linking onto memory consolidation. Indeed, pre-learning injections of AMPAR cross-linkers did not impact animal performance on day 1 (pre-learning; Fig. 1d, left and 1e, left), but rather on the following day, characterized by mice's choices returning to random level (Fig. 1d, left and 1e, right). A similar effect was observed when injections were performed immediately after session #4 (pre-rest; Fig. 1d, middle and 1f), but not if done before the test performed on day 2 (pre-test; Fig. 1d, right and 1g). Thus, the results indicated that memory retrieval was not impacted by AMPAR cross-linking, while pointing that the AMPARM-dependent process occurs during the resting period that is thought to support memory consolidation.

An important question to be addressed is the origin of performance deficits observed on day 2 in pre-learning and pre-rest injected animals. Indeed, an increase in the number or errors can be characterized by various origins such as disorientation, disengagement, bad animal state, up to the complete forgetting of the DSA rule. To further distinguish between these options, we thoroughly analyzed animal behavior during the acquisition of the DSA rule (Fig. 2). As in the other groups using T and/or Y mazes to test for mice cognitive abilities[17,18], we noticed that runs can be separated into two groups: those in which the animals were running in the maze with almost constant speed and those in which hesitation can be observed at the crossing point, with significant changes in head orientation and speed, called vicarious trial and error runs or VTE runs (Fig. 2a) that are predictive of accurate good choice, as testified by the difference in the error rates in VTE and non-VTE runs (Fig. 2a, right). Interestingly, probably because DSA rule was not yet clear, animals exhibited more no-VTE runs at the beginning of the learning day (Fig. 2b).

Next, to get insights in the origin of the performance loss of pre-learning treated animals, we examined the occurrence and choice quality of VTE and no-VTE runs (Fig. 2b–e). Surprisingly, the lack of performance of pre-learning IgG-injected animals was associated with reinstatement of initial values, suggesting an apparent amnesia of DSA rule (Fig. 2c–e): (i) in IgG-treated mice, the occurrence of no-VTE runs in session #5 was similar to its initial session #1 value (Fig. 2b, c), as if the animals had to re-acquire DSA rule, rather than being unable to apply the beforehand encoded one; (ii) the error rate of VTE runs in session #5 also increased, returning to the level observed in session #1 (Fig. 2d), suggesting that when attempting to apply the rule, animals behaves randomly, as initially (Fig. 2e, bottom) and (iii) whereas the no-VTE runs performance improved with training and progressively diverged from random choices, they also returned back to their initial values in session #5 in IgG-injected mice (Fig. 2e, top). Importantly, on day 1, performance during VTE runs evolved significantly in IgG- and control-injected animals, suggesting that DSA rule encoding processes are ongoing normally, even in the absence of AMPARM (Fig. 2b–e and Supplementary Fig. 1). Based on this, we propose that a hippocampal AMPARM-dependent mechanism is involved in consolidating memory during resting periods following training, and that dorsal hippocampus (dHPC) AMPAR immobilization leads to a total forgetting of the acquired DSA rule rather than an incorrect encoding or execution of it.

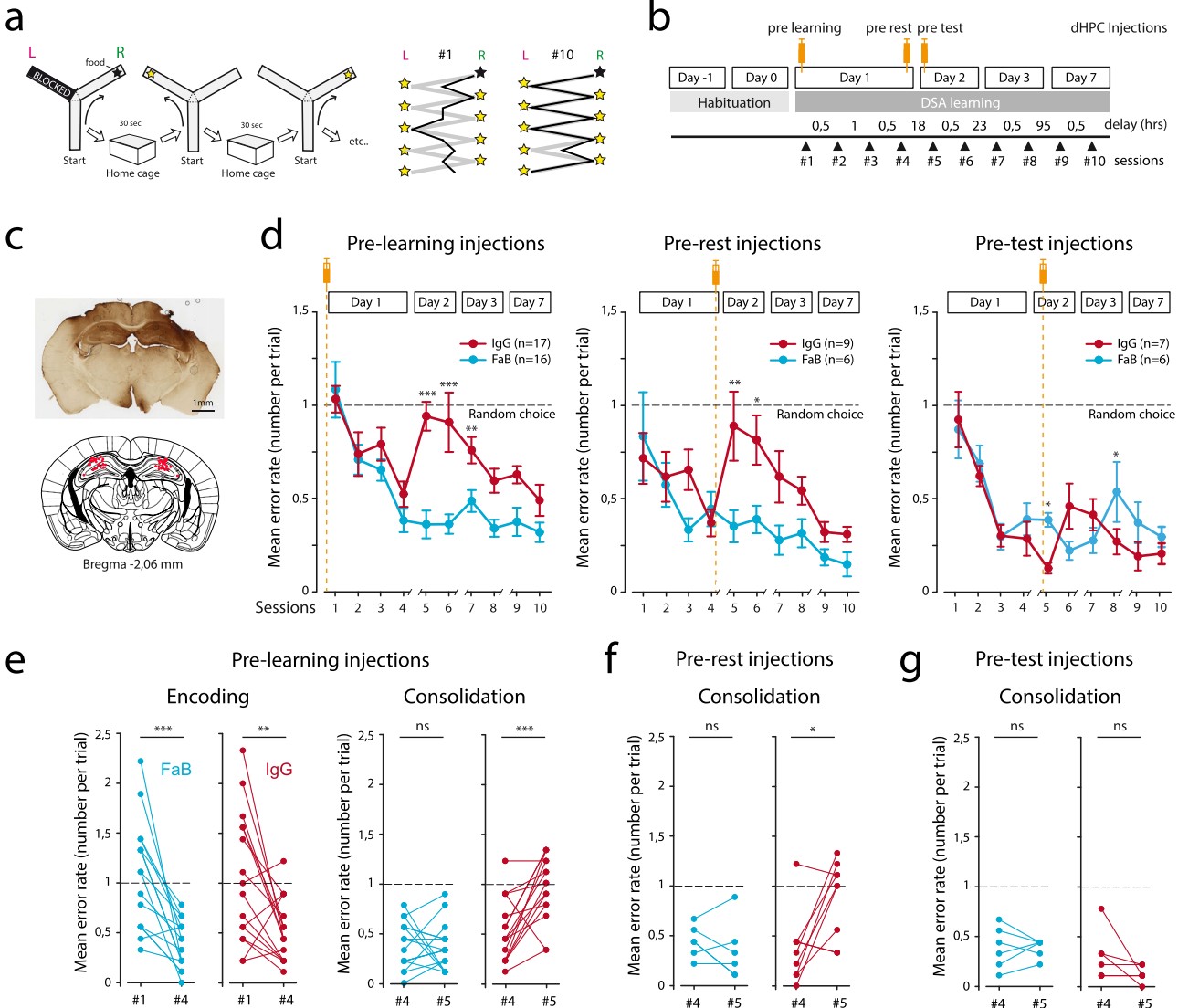

**Fig. 1 | AMPAR surface mobility in the dorsal hippocampus is necessary for the consolidation of a delayed spatial alternation rule. a** Schematic of the Delayed Spatial Alternation task. In each session, after an initial forced choice, nine food-rewarded positions−called trials, alternatively right and left−are set. Up to five error runs are permitted per trial before the animal is forced to enter in the rewarded arm. In between runs, the animal is positioned in its home cage for 30 s. After 10 training sessions (session #1 to #10) allocated within a week, animals are alternating almost perfectly (right panels). **b** Intracerebral bilateral injections in the dorsal hippocampus were performed at three different time points of the DSA training: pre-learning (before session #1), pre-rest (after session #4) and pre-test (before session #5) (orange syringes). **c**: Two cannulas were implanted above the dHPC, and anti-GluA2-IgGs or control compounds were injected. Template is from "the mouse brain" Paxinos and Franklin. **d** Behavioral results obtained in the various cohorts expressed as mean error rates. Injections of anti-GluA2 bivalent IgGs (red) or monovalent Fabs (blue) were performed as indicated. A two-way ANOVA was performed to analyze the effect of time and treatment on error rates. Statistically significant interaction between time and treatment was found for pre-learning injections ($F(9,308) = 2.204$, $p = 0.022$), but not for pre-rest ($F(9,130) = 1.547$, $p = 0.138$) and pre-test ($F(9,110) = 1.632$, $p = 0.115$) injections. FaB vs IgG pairwise multiple comparisons were done using the Holm−Sidak method. *$p < 0.05$, **$p < 0.01$, ***$p < 0.001$. **e** Single animal data are shown for crucial behavioral steps. Memory encoding (left) is achieved within the first day, the error rate being minimal at session #4 (**d**). Error rates between session #4 and session #5 are similar in FaB-injected control animals, but returned to chance level in IgG-injected mice. **f**, **g** Same presentation as in (**e**), but for consolidation−session #4 vs session #5−in pre-rest (**f**) and pre-test (**g**) injected cohorts. Paired $t$-tests were used. In case that sample distribution was not normal−after the Shapiro−Wilk test−a Wilcoxon ranked test was used. *$p < 0.05$, **$p < 0.01$, ***$p < 0.001$. $n$ represents the number of injected animals. Data are presented as mean values ± SEM.

## AMPAR immobilization in the dorsal hippocampus impairs ripple physiology during slow-wave sleep

Hippocampal ripples are fast oscillations that develop during slow wave sleep (SWS) phase and that are considered offline replays of immediately preceding experiences[6,8,19]. They are generated in CA2/CA3 regions of the hippocampus, and propagate in CA1 before broadcasting to cortical regions[9,20]. Interestingly, their interplay with immediately preceding synaptic tagging is unknown, even if specific downscaling and NMDAR-dependent synapse refinement have been reported in in situ preparations[11].

Thus, we wanted to examine the impact of IgG treatment and/or DSA learning on dHPC ripples (Fig. 3). To achieve that, animals were implanted bilaterally with wire bundles medially to injection cannula (Supplementary Fig. 2a). dHPC Local Field Potentials (LFPs) were recorded for 3 h immediately following Y-maze habituation ("habituation" in Day 1 or D-1) or after the first 4 DSA sessions ("DSA" in Day 1 or D1, Fig. 3). At first, we separated awake and resting/sleeping state in the home cage using animal tracking (mobility, Fig. 3a, top and Supplementary Fig. 2c). Then, slow wave sleep (SWS) and rapid eye movement (REM) sleeping phases were separated using a Theta/Delta

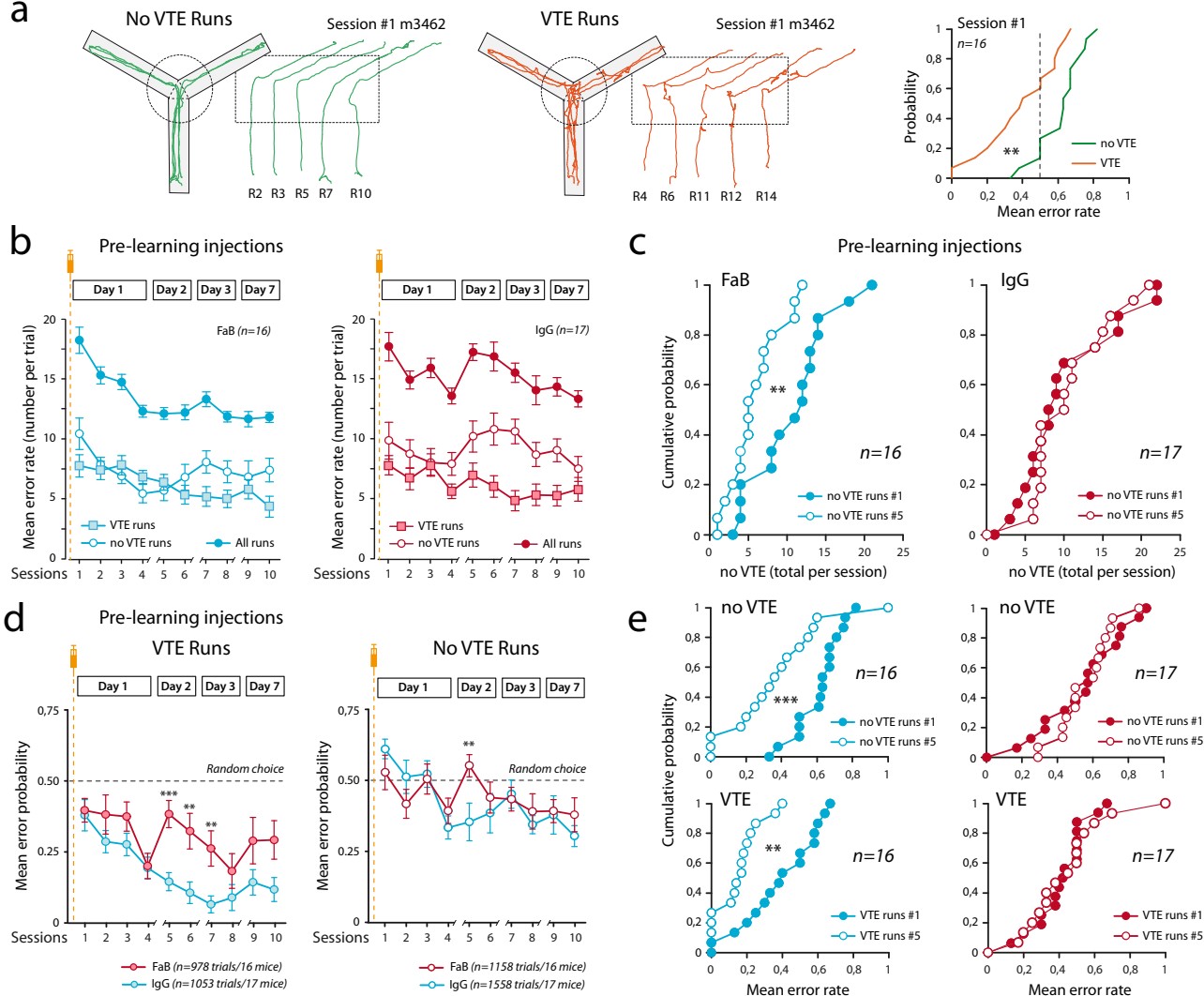

**Fig. 2 | Immobilization of AMPAR in the dorsal hippocampus led to complete forgetting of the acquired DSA rule. a** DSA runs can be separated in two groups according to animal hesitation at the middle of the maze. As defined by ref. 18, vicarious trial and errors (VTE) behavior indicates mouse cognitive engagement in the task. Left: examples of no-VTE (green) and VTE runs for mouse 3462. Right: animals performance at session #1 for VTE and no-VTE runs, Mann−Whitney rank sum test, $p = 0.002$. $n$ are independent animals. **b** For pre-leaning injections cohorts, number of VTE and no-VTE runs was analyzed along the DSA sessions. Note that upon IgG injections, the number of no-VTE runs at session #5 was similar to session #1. $n$ are independent animals. Data are presented as mean values ± SEM. **c** Cumulative single animal data for no-VTE run numbers in session #1 and session #5. $t$-tests were used to compare the evolution of no-VTE runs occurrence between sessions #1 and #5. FaB: $p = 0.005$; IgG, $p = 0.862$. $n$ are independent animals. **d** Choice acuteness during VTE and no-VTE runs was analyzed along the DSA

sessions in the pre-learning injection cohorts. The lack of animal performance at session #5 (see Fig. 1d) is associated with a decrease in VTE run accuracy that returned to its initial value in session #1. A milder effect is observed for no-VTE runs. A two-way ANOVA was performed to analyze the effect of time and treatment on error rates for VTE (left) and non-VTE (Right) runs. No statistically significant interaction between time and treatment was found for either VTE runs ($F(9,294) = 1.463$, $p = 0.161$) or no-VTE runs ($F(9,304) = 1.327$, $p = 0.222$). FaB vs IgG pairwise multiple comparison procedures were done using the Holm−Sidak method. ** $p < 0.01$, *** $p < 0.001$. Data are presented as mean values ± SEM. **e** Same presentation as in (**c**) for choice accuracy for VTE and no-VTE run numbers in session #1 and session #5. $t$-tests were used. In case sample distribution was not normal−after the Shapiro−Wilk test−a Mann−Whitney rank sum test was used. ** $p < 0.01$, *** $p < 0.001$. $n$ are independent animals.

ratio defined on hippocampal LFP spectra (Fig. 3a, middle and Supplementary Fig. 2c). REM periods host robust theta oscillations absent in SWS periods, that are characterized by pronounced Delta waves (for a typical example see Supplementary Fig. 2c). Importantly, as expected, SWS periods correlate nicely with the occurrence of hippocampal ripples (see methods, Fig. 3a, bottom and Supplementary Fig. 2c).

Then we tested if the DSA protocol and AMPAR cross-linking were leading to alterations in ripple frequency and amplitudes (Fig. 3b–f). At first, we tested if our recordings were stable over time. Indeed, and not surprisingly, neither the amplitude nor the frequency of detected ripples differed between two basal consecutive days (Controls D-2 and D-1 recordings, Fig. 3c). Some reports described that spatial learning or

retrieval was leading to an increase in dHPC ripple frequency[21]. However, no noticeable changes in recorded ripples were observed in animals submitted to DSA learning, and injected with control constructs or non-injected (see specific mentions in Fig. 3d). We next tested if the blockade of AMPARM in the dHPC was perturbing ripple physiology in DSA-trained animals (Fig. 3e) or non-trained mice (Fig. 3f). In both cases, IgG injections were done at pre-learning time, i.e., several hours before the recorded resting periods, the only difference being that non-trained animals are solely positioned in the maze for a similar time duration, but with no specific rule to learn (see methods). Surprisingly, we detected a significant impact of IgG injections on ripple amplitude and frequency that were significantly

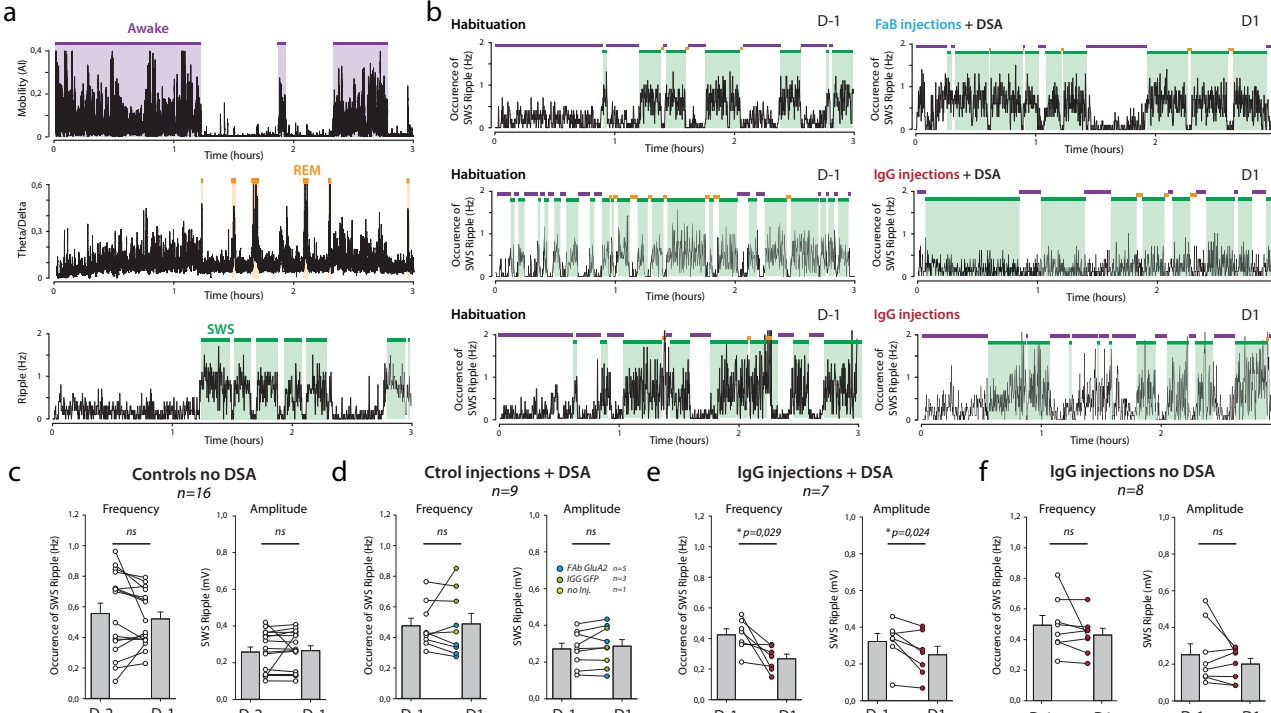

**Fig. 3 | Immobilization of AMPAR in the dorsal hippocampus led to learning-dependent impairment of ripple activities. a** During resting periods in the home cage, animal tracking and dHPC electrophysiological LFP recordings were performed to temporally define three animal states: awake state is defined by animal mobility, whereas resting states were separated in rapid eye movement (REM) and slow wave sleep (SWS) phases, according to the Delta/Theta ratio of dHPC LFPs. LFP signals were also filtered at 150–250 Hz to extract ripples, the frequency which is closely correlated with SWS periods[6]. **b** Bilateral dHPC LFPs were recorded for 3 h resting periods before (habituation, D-1) or after DSA encoding (after session #4, D1). Typical examples of ripple frequency in pre-learning FaB (Top) and IgG-injected (middle) animals, or in IgG-injected animals that were not subjected to DSA learning (bottom). Note the decrease in SWS-ripple frequency in D1 of the IgG-injected DSA-trained animal. **c–f** Amplitude and frequency of ripples during SWS periods were extracted in D-2, D-1 or D-1 resting periods (as indicated) in four different groups: **c** day-to-day recordings with no injection nor DSA learning, **d** before and after control drug injections and DSA learning, **e** before and after IgG injections and DSA learning, **f** before and after control IgG injections but no DSA learning. Paired t-tests were used. In case that sample distribution was not normal–after the Shapiro–Wilk test–a Wilcoxon ranked test was used. ns not significant, *$p < 0.05$, **$p < 0.01$, ***$p < 0.001$. n is the number of recorded animals. Data are presented as mean values ± SEM.

decreased, but this effect was only observed when learning was present (compare Fig. 3e, f). Because ripple inactivation during SWS has been proven to impair spatial memory consolidation[22], this decrease in ripple content may explain the lack of consolidation observed in pre-learning IgG-injected animals (Fig. 1d). Thus, our data indicate that post-learning AMPARM-dependent plasticity events in the dHPC support the genesis and strength of hippocampal ripples.

## AMPARM-dependent plasticity at CA3 recurrent synapses support ripple activity in situ

Most of what we know about cellular and synaptic contribution to ripple physiology comes from acute in situ preparations in which ripple-like oscillations are spontaneously generated[10,12,13]. While certain aspects of in vivo ripples are absent, such as their cognitive content[6], in situ ripples are still thought to recapitulate most of the in vivo ripple properties and link with in vivo experience[6,11].

Here, we wanted to address if an interplay exists between AMPARM-dependent plasticity and ripple physiology. For that, we setup and used in situ hippocampal preparations exhibiting spontaneously ripples and combined them with synaptic "tagging" by inducing LTP at CA3→CA1, CA3→CA3 and DG→CA3 synaptic contacts with or without the presence of AMPAR cross-linkers (Figs. 4 and 5). In optimized in situ preparations[13], ripple-like activities–here called SPW-Rs–can be stably and robustly recorded using field recording pipettes positioned in the CA3 and CA1 regions (Fig. 4a–e). To be included, SPW-Rs recordings have to have stable occurrence frequency, showing co-detected CA3 and CA1 events, present a constant delay between

CA3 (first) and CA1 (delayed) responses (Fig. 4c, middle), and have a good amplitude matching between both signals (Fig. 4b, c, right). Some other criteria were eventually respected when present: (i) the signal polarity in the CA1 region was dependent of the recording location: positive in the *stratum pyramidale*, and negative in the *stratum radiatum*, confirming that incoming CA3 activities were generating a significant synaptic field response in CA1 (Supplementary Fig. 3a), (ii) both evoked and spontaneous SPW-Rs eventually engaged CA3 unitary activities (Supplementary Fig. 3b), (iii) when tested, stimulations in the CA1 *stratum radiatum* that generated SPW-Rs were interfering with spontaneous SPW-Rs, generating a refractory period (Supplementary Fig. 3c). Importantly, after a 20-min period in the recording chamber, all parameters were stable for more than an hour, allowing the combination of SPW-Rs recordings with high frequency stimulation (HFS) application and/or pharmacological manipulations (Figs. 4 and 5).

We previously showed that AMPAR immobilization at the neuronal surface in the CA1 region impaired LTP expression at CA3→CA1 synapses[15]. We first aimed to reproduce and extend this finding to other synapses eliciting post-synaptic LTP expression. Interestingly, in the CA3 region, pyramidal neurons receive two major excitatory afferent that are intrinsically different. Mossy fibers originating in the dentate gyrus generate "detonating" synapses expressing a huge rate-dependent facilitation that can be prolongated by a sustained potentiation of presynaptic origin (presynaptic release probability increase[23]). In contrast, recurrent synapses emitted by distant or neighboring CA3 pyramidal cells are classical Hebbian synapses,

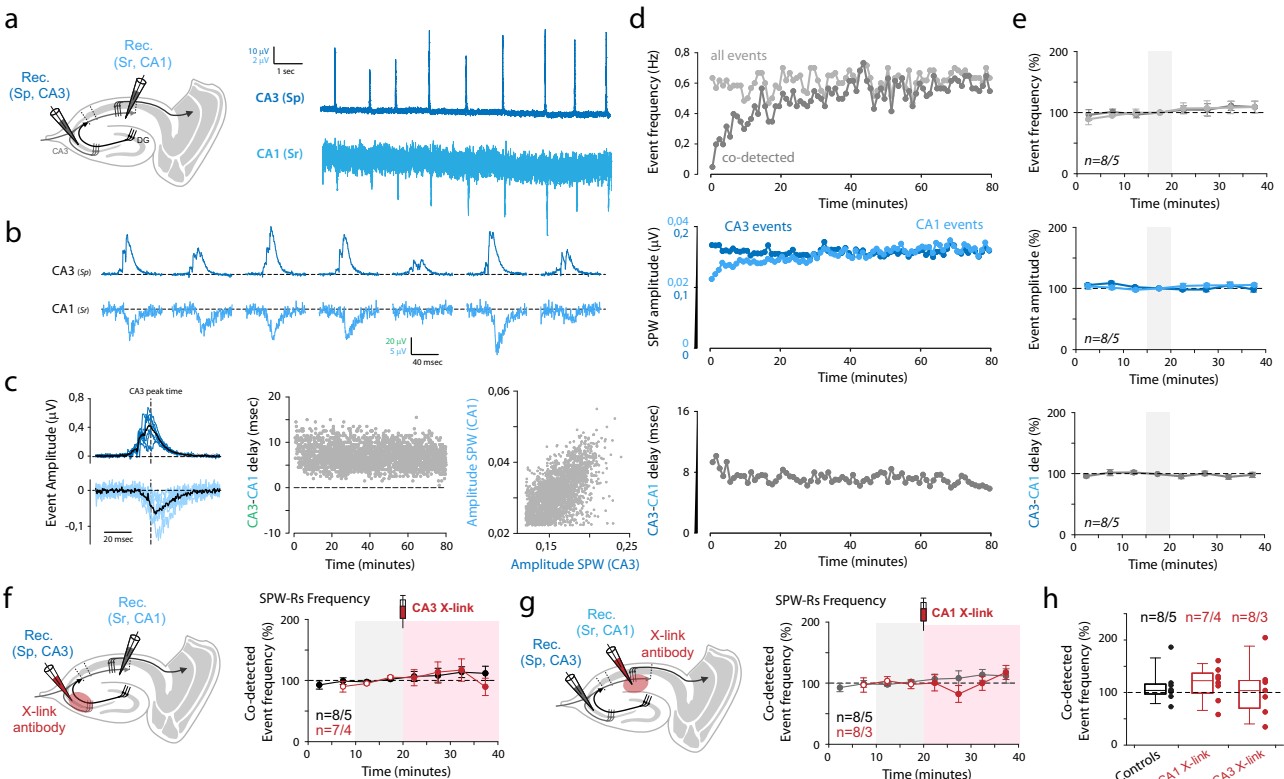

**Fig. 4 | Immobilization of AMPAR in the dorsal hippocampus did not affect spontaneous ripple activities in naive in situ preparations. a** Spontaneous sharp waves events (SPW-Rs) are recorded in fresh in situ hippocampal preparations (see methods) using extracellular field electrodes positioned in the *stratum radiatum* (Sr) or the *stratum pyramidale* (Sp) of CA3 or CA1 regions. **b** Examples of recorded events. **c** CA1/CA3 simultaneously recorded events showed a significant delay (left), and correlated amplitudes (right) suggesting their propagation from CA3 to the CA1 area. **d**, **e** Single example (**d**) and averaged (**e**, *n* = 7 independent experiments, presented as mean values ± SEM) measures showing the stability of SPW-Rs frequency (top), amplitudes (middle) and delay (bottom) with time. All pharmacological experiments start after respecting a 20-min period required for SPW-Rs

stabilization. **f**, **g** Effect of AMPAR X-linking on spontaneous SPW-Rs was tested in in situ preparations by pressure injection of anti-GluA2 IgGs. Left: schematic of the experiment. Right: time course of co-detected SPW-Rs frequency. The "pink" area indicates that events are recorded in the IgG-injected area (for IgG condition, red dots). Control condition−no injection−is shown in black dots. **h** All single experiments and average values after 20 min, in control or after IgG injections. *t*-tests were used, but failed in finding any difference between groups (CA3 X-link vs ctrl: *p* = 0.756; CA1 X-link vs ctrl: *p* = 0.803). The box plot minima and maxima are 25–75% with a center at the median. Whiskers are at 10 and 90%. **f–h:** *n* = X/Y with X biologically independent slices examined over Y independent animals. Data are presented as mean values ± SEM.

expressing post-synaptic LTP[23]. We thus tested the impact of AMPAR cross-linking on LTP in our in situ "SPW-Rs" preparation (Supplementary Fig. 4). Not surprisingly, we observed that AMPAR X-linking led to an absence of LTP at synapses post-synaptic to CA3 axons (CA3→CA1 and CA3→CA3) but not at DG→CA3 projections, that were solely affected by PKA blockade using Rp-cAMP preincubations (Supplementary Fig. 4).

In the absence of learning-associated synaptic tagging, AMPAR immobilization did not lead to changes in ripple frequency or amplitude (Fig. 3f). Acute slice preparations from naive mice are often used as models to study molecular and cellular mechanisms of LTP induction and expression at hippocampal synapses[24]. It is then often considered that in naive mice, no significant synaptic tagging− endogenously triggered LTP−is present. We thus tested the effect of AMPAR cross-linking in SPW-Rs containing naive preparations by locally infusing anti-GluA2 IgGs in CA1 or CA3 *stratum radiata* (Fig. 4f−h). Notably, the efficacy of this injection procedure on LTP expression was previously validated in CA1[15], and reiterated here in CA1 and CA3 regions (Supplementary Fig. 4). When compared to basal conditions, these injections had no effect on SPW-Rs frequency or amplitude (Fig. 4f−h and Supplementary Fig. 5). The local effect of IgG injection on amplitude being attributable to the one/two pipette(s) procedure (Supplementary Fig. 5). Therefore, in line with the absence of effect of AMPAR immobilization on basal synaptic transmission[15], and in our in vivo data obtained in naive mice (Fig. 3), our in situ results

suggest that basal SPW-Rs did not rely on AMPARM in the absence of specific synaptic tagging.

Next, we wanted to test if synaptic tagging−here generated by HFS applications enabling LTP induction (Supplementary Fig. 4)− could modulate SPW-Rs frequency (Fig. 5). Interestingly, CA1 HFS stimulations did not impact SPW-Rs frequency (Fig. 5a, c), indicating that synaptic strength at CA3→CA1 synapses may not be a determinant of SPW-Rs genesis. Strikingly, the same procedure applied at CA3→CA3 recurrent synapses led to a strong increase in SPW-Rs frequency (Fig. 5b, c), prominent in case of low basal SPW-Rs frequency (Fig. 5c). Furthermore, we observed that the effect of HFS on synaptic strength and SPW-Rs frequency seems to have different time courses, the increase in evoked fEPSP amplitude being detectable as early as in the 0−5 min post-tetanic period, whereas the effect on SPW-Rs frequency was not yet present (Fig. 5b2, bottom, 5c). This possibly reflects an ongoing development of synaptic inputs potentiation, along with a progressive rise in CA3 cells excitability. Thus, these results suggested that the reinforcement of CA3→CA3 recurrent synapses increases CA3 region excitability and promotes the generation of ripples.

Finally, we tested if AMPAR immobilization, in addition to abolishing LTP at CA3→CA3 synapses, would also perturb SPW-Rs modulations by CA3 HFS. We applied HFS-CA3 stimulations in SPW-Rs expressing slices in which local infusions of anti-GluA2 IgGs were performed in the CA3 region (Fig. 5d−e). Under AMPAR

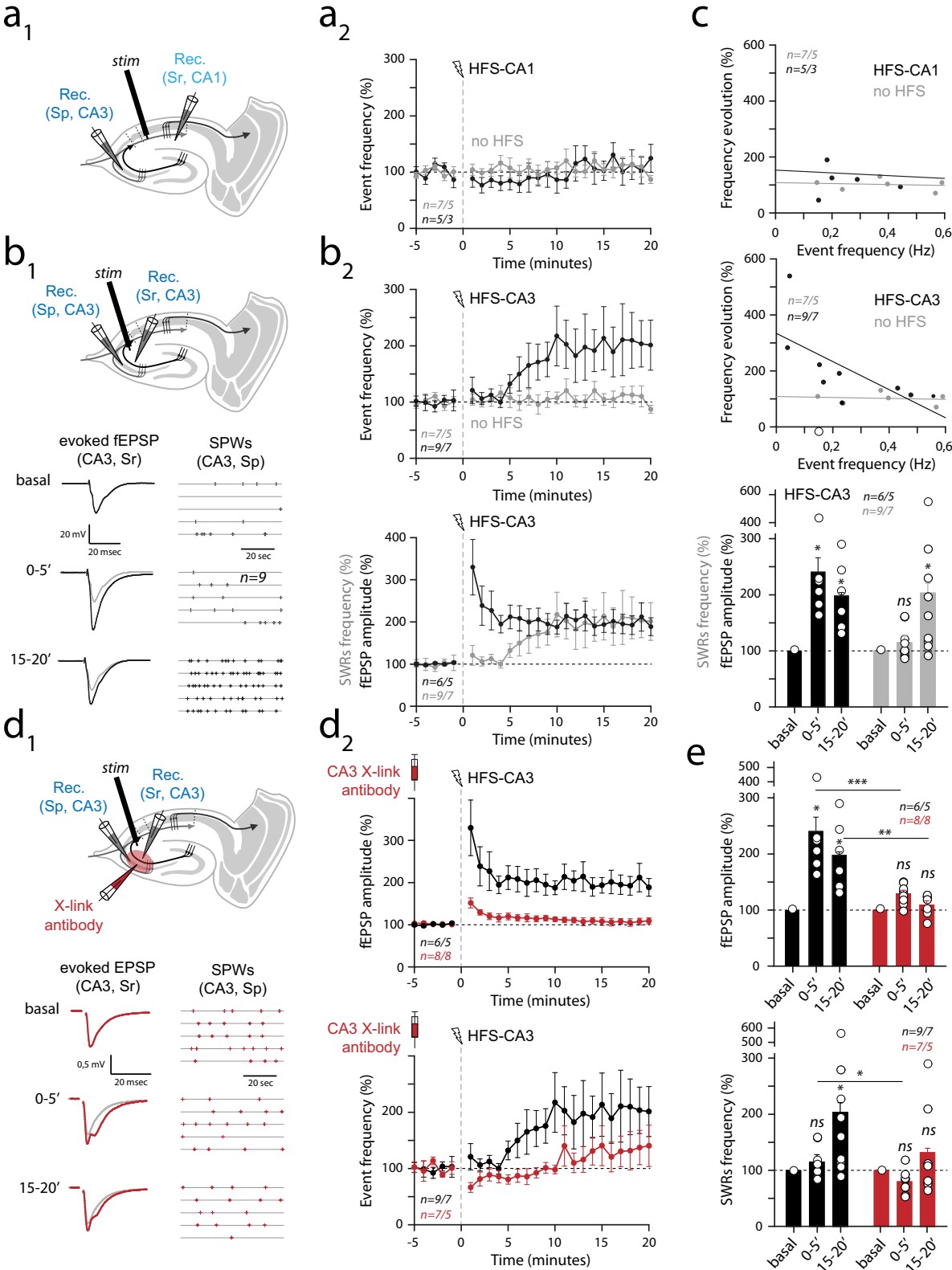

immobilization, the HFS-associated effect onto SPW-Rs frequency was minimized (Fig. 5d–e), suggesting that their physiology depends on AMPARM-dependent CA3 recurrent synaptic strength. Of note, a contribution of synaptic potentiation at DG→CA3 synaptic inputs is unlikely to contribute to the HFS effect. indeed, we systematically tested for 1 Hz frequency facilitation in our evoked synaptic responses in CA3-sr, retaining only experiments with negligible facilitation, and

HFS-triggered plasticity at these synapses is insensitive to AMPAR cross-linking (Supplementary Fig. 4). Thus, we propose that AMPARM-dependent LTP at CA3 recurrent synapses positively controls ripple activity in situ. Together with our in vivo data, our findings suggest that CA3→CA3 synaptic tagging may be triggered during DSA learning, which is important for ripple-mediated consolidation occurring during upcoming sleep phases.

**Fig. 5 | Interplay between plasticity induction, spontaneous SPW-Rs and AMPAR mobility in hippocampus in situ. a1** Spontaneous sharp wave ripples were recorded in hippocampal acute slices. After stabilization, application of HFS in the *stratum radiatum* of CA1 (CA1-sr) was applied to induce LTP at CA3→CA1 synapses (see Supplementary Fig. 4). **a2** No major impact of CA1 HFS onto SPW-Rs frequency or amplitude was observed. Data are presented as mean values ± SEM. **b1** Top: same presentation as in (**a1**) for HFS application in CA3-sr. Bottom: typical example of the effect of CA3 HFS on evoked CA3 responses (left, gray line is basal trace), and frequency of spontaneous SPW-Rs (right). **b2** Top: same presentation as in (**a2**). Data are presented as mean values ± SEM. Note the increase in SPW-Rs frequency and amplitude after HFS application. Bottom: a significant delay exists between synaptic potentiation (evoked fEPSPs, black dots) and SPW-Rs frequency increase (gray dots). **c** Results presented in (**a**) and (**b**) are summarized. Data are presented as mean values ± SEM. Top: the effect of CA3 HFS (black dots) depends on the initial SPW-Rs frequency. SPW-Rs frequency evolution with time did not depend on initial frequency (gray dots). Bottom: relative fEPSP amplitude and

SPW-Rs frequency 0–5 and 15–20 min after CA3 and HFS applications. Paired *t*-tests were used. In case that sample distribution was not normal—after the Shapiro–Wilk test—a Wilcoxon ranked test was used. ns not significant, *$p < 0.05$. (**d1/d2**) Same presentation as in (**b1/b2**). Data are presented as mean values ± SEM. 10 min before electrophysiological recordings, IgG injections were performed in the recorded CA3 region (CA3 X-link antibody). Note that all effect triggered by CA3 HFS in control conditions are absent in the presence of IgG. Number of recordings is indicated. **e** Same presentation as in (**c**). For intra-group time course comparisons, paired *t*-tests were used. In case that sample distribution was not normal—after the Shapiro–Wilk test—a Wilcoxon ranked test was used. ns not significant, *$p < 0.05$, **$p < 0.01$. For comparisons of similar timing between groups, *t*-tests were used. In case sample distribution was not normal—after the Shapiro–Wilk test—a Mann–Whitney rank sum test was used. *$p < 0.05$, **$p < 0.01$, ***$p < 0.001$. $n = X/Y$ with X biologically independent slices examined over Y independent animals. Data are presented as mean values ± SEM.

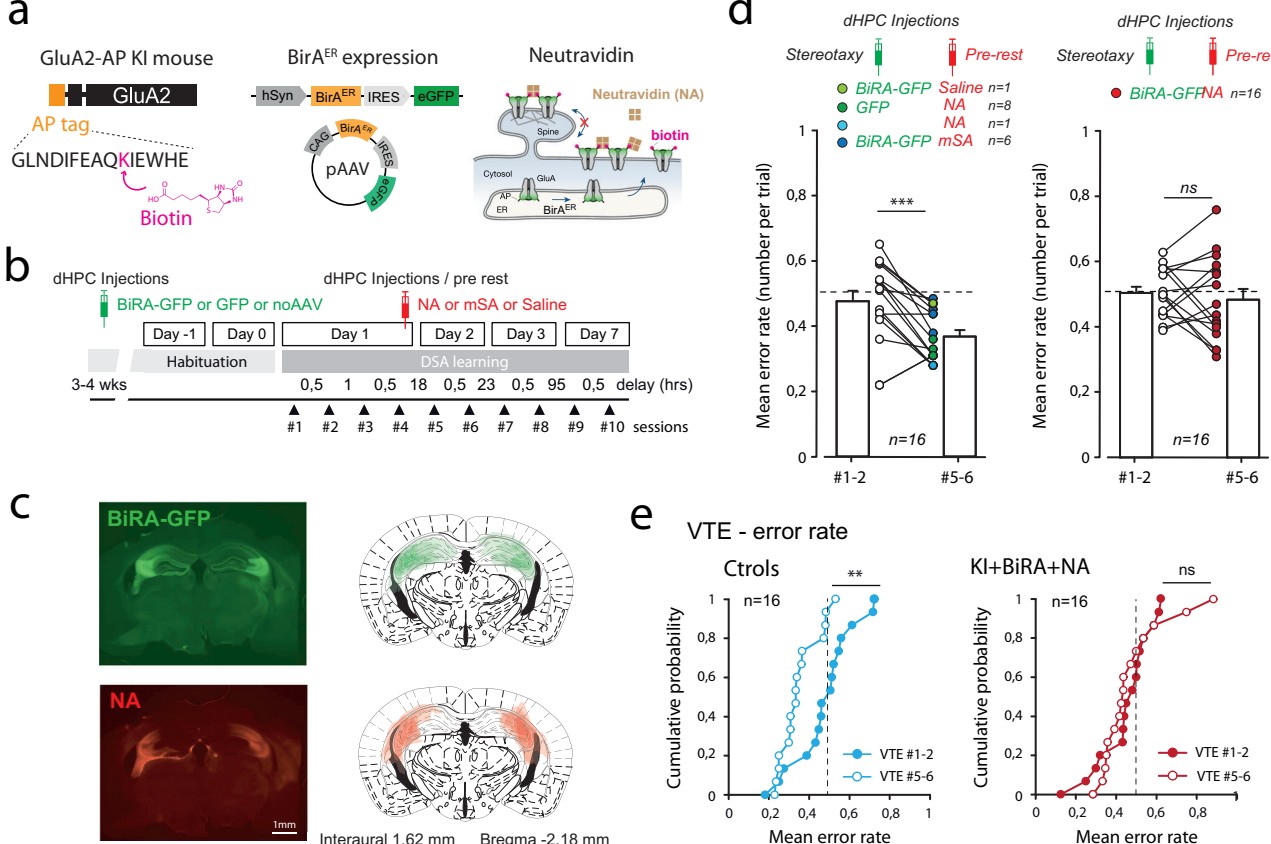

**Fig. 6 | An alternative AMPAR X-linking strategy allowing a better targeting of the CA3 area also induced complete forgetting of DSA rule. a** We recently developed a new strategy for AMPAR X-linking. Knock-in mice expressing endogenous AP-tagged GluA2 AMPAR subunits can be biotinylated in the presence of BiRA[ER], and once exported to the cell surface can be immobilized in the presence of external neutravidin (NA, cross-linking condition). **b** Similar in vivo pharmacological experiments as in Fig. 1d were performed, combining early stereotaxic dHPC injections of AAV-BiRA-GFP or AAV-GFP, and pre-rest injections of saline, mSA or NA. **c** Histological controls for the mSA and NA staining on top of the AAV-GFP

expression. The combination of both injections better restrict AMPAR immobilization to the CA3 area. Template is from "the mouse brain" Paxinos and Franklin. **d** Mean error rates were compared between sessions #1–2 and sessions #5–6 to evaluate the retention of the DSA rule upon various pharmacological treatments (as indicated by color coding). Paired *t*-tests were used. ns not significant, ***$p < 0.001$. Data are presented as mean values ± SEM. **e** The error rate in VTE runs was reported in sessions #1–2 (filled dots) and 5–6 (empty dots) for control (left) and cross-link (right) groups. *t*-tests were used. ns not significant. **$p < 0.01$. $n$ = biologically independent animals.

## AMPAR mobility in CA3 area is necessary for memory consolidation

Based on our in situ data, we next wanted to restrict AMPAR cross-linking in the CA3 area and evaluate if local impairment of CA3 plasticity would be sufficient in affecting memory consolidation and ripple physiology. However, the antibody-based AMPAR cross-linker strategy lacks spatial and temporal resolution, as in vivo injections often cover

large parts of the dHPC[15] (Fig. 1). In addition, antibody-associated changes of AMPAR composition[25] may be present and can lead to misinterpretation of the data (see discussion). Thus, we used a recently developed approach to cross-link endogenous GluA2-containing AMPAR using biotin/streptavidin complexes[16] (Fig. 6a). In Knock-in mice expressing AP-tagged GluA2 subunits, the presence of an exogenous enzyme—BiRA[ER], brought by viral infections—allow the

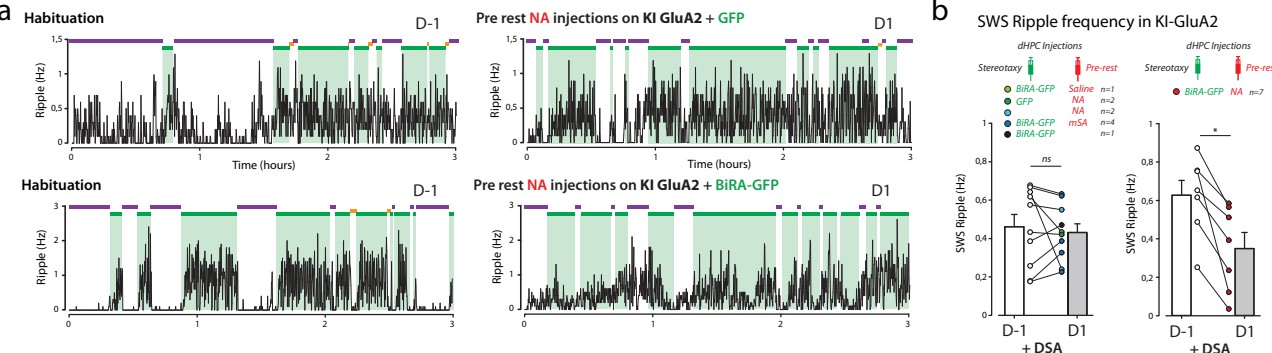

**Fig. 7 | An alternative AMPAR X-linking strategy allowing a better targeting to the CA3 area affected SWS ripple activity. a** Same presentation as in Fig. 3b. Bilateral dHPC LFPs were recorded for 3 h resting periods before (habituation, D-1) or after DSA encoding (after sessions session #4, D1). Typical examples of ripple frequency in control (top and bottom) and X-linking (middle) conditions. Note the decrease in SWS-ripple frequency in D1 of the KI-BiRA-NA DSA-trained animal. **b** Same presentation as in Fig. 3c. Data are presented as mean values ± SEM. SWS ripple frequency was analyzed and reported for single experiments (dots) and averaged (bars). Conditions and animal numbers are indicated. Paired t-tests were used. ns not significant, *p < 0.05. n = X biologically independent animals.

biotinylation of GluA2-containing AMPAR, that can be cross-linked in the presence of tetravalent neutravidin added in the extracellular space (Fig. 6a). This cross-linking approach that has been validated in vitro and in vivo[16] and among other advantages improve spatial resolution through a combination of viral expression and drug delivery via intracerebral cannula (Fig. 6b). We first validated our capacity to target specifically the CA3 area by infusing NA-texasRed (red-tagged tetravalent neutravidin) through cannula implanted above the CA3 regions of BIRA-expressing mice (Fig. 6c). Indeed, red labeling was almost restricted to the CA3 region, in a subpart of the green expressing region (Fig. 6c). Then, we tested the DSA rule expression on day 2 from mice with pre-rest CA3 cross-linking and various control conditions. This time point was chosen because the time course of NA action is not yet ascertained (Fig. 6b). Furthermore, as GluA2 KI animals were slow in learning DSA rule, we mixed sessions #1–2 and #5–6 to get more robust behavioral outcomes. When compared to initial scores (sessions #1–2), error rate of control animals was significantly lower in day 2 tests (sessions #5–6, Fig. 6d), indicating that encoding and consolidation of DSA rule were successfully achieved in the GluA2-AP KI mice. In contrast, error rates at these two time points were close to random values in X-linking conditions (Fig. 6d) a phenotype that is again accompanied by an apparent forgetting of the DSA rule. Indeed, the accuracy of VTE runs improved in control mice, but remained unchanged in the X-linking conditions (Fig. 6e).

To confirm that the lack of consolidation is associated with impairment of ripples physiology, we combined this novel cross-linking method with dHPC recordings using the same recording methodology as for IgG experiments (Fig. 3). Ripples occurring during SWS were extracted and counted in habituation (D-1) and after DSA (D1) sessions (Fig. 7). As the last recording started 1 h after drug delivery, it was important to control for unspecific effects of drug actions. Of note, NA application on GFP only and saline or mSA delivery on BiRA-expressing dHPCs were not leading to changes in ripple frequency, as for DSA consolidation (Fig. 6). However, NA delivery in BiRA-expressing CA3 areas was associated with a pronounced decrease in ripple frequency, reminding the effect observed upon pre-learning and pre-rest IgG injections (Fig. 3). Therefore, through utilization of two different cross-linking approaches, we demonstrated that AMPARM in the CA3 region is necessary for memory consolidation and support ripple physiology during slow wave sleep.

## Discussion

In order to understand the link between synaptic plasticity and learning and memory it is essential to analyze separately the various phases of memory encoding, consolidation and retrieval separately, and to use specific tools that disturb plasticity without affecting basal synaptic transmission. With this in mind, we developed molecular strategies that can be used in vivo to address these issues. We tested two different methods to impair post-synaptic long-term potentiation in the dorsal hippocampus, and uncovered that during the process of acquiring a spatially guided appetitive rule, AMPAR mobility was necessary during the consolidation phase and was an important physiological mechanism that supports ripple activity consecutive to new rule encoding. We concluded from our in situ experiments that an interplay between AMPARM, LTP at CA3→CA3 recurrent synapses and ripple physiology emerged, allowing us to propose a model (Fig. 8) in which rule encoding, possibly through synaptic tagging, conditions/organizes subsequent ripple activity that will develop during rest to consolidate memory. This would require the occurrence of AMPARM-dependent LTP at CA3 recurrent synapses, as the immobilization of AMPAR in CA3 in vivo during consolidation leads to a learning-dependent loss of ripple activity and memory forgetting. Thus, we bring a novel mechanism by which synaptic plasticity contributes to learning and memory.

### Controls for AMPAR X-linking strategies

Some antibodies against GluA2 subunits have been reported to modify AMPAR composition within several hours[25], a time window that may correspond to the consolidation phase of the DSA rule in the case of pre-learning injections. Thus, one could attribute the observed effects onto consolidation to significant changes in hippocampal connectivity and activity consecutive to the replacement of GluA2-containing heteromers by GluA1 homomers. However, we observed a very similar effect on animal performance regardless of whether intracerebral IgGs were infused before or after DSA rule encoding (Fig. 1d–f), a specific time at which the IgG-dependent changes in AMPAR composition might not have yet occurred. The same reasoning can be applied to the ripple activity in Fig. 3. The application of pre-learning IgG in the absence of DSA learning did not impact the network capacity in generating ripples. In fact, the impact of both strategies appears to be prominent when efficient cross-link is present at the time of offline rule consolidation.

A methodological concern regarding in vivo strategies and ripple detection is that local delivery of drugs affects the electrophysiological recordings due to volume injection, an issue that was identified in our in situ experiments (Supplementary Fig. 5). If this occurred in vivo, a global decrease in LFP amplitude caused by tissue movements would have potentially affected our capacity to detect ripples after injection. At first, we tried to minimize this effect by decreasing the injection speed (see methods) and temporally separating the injection from the

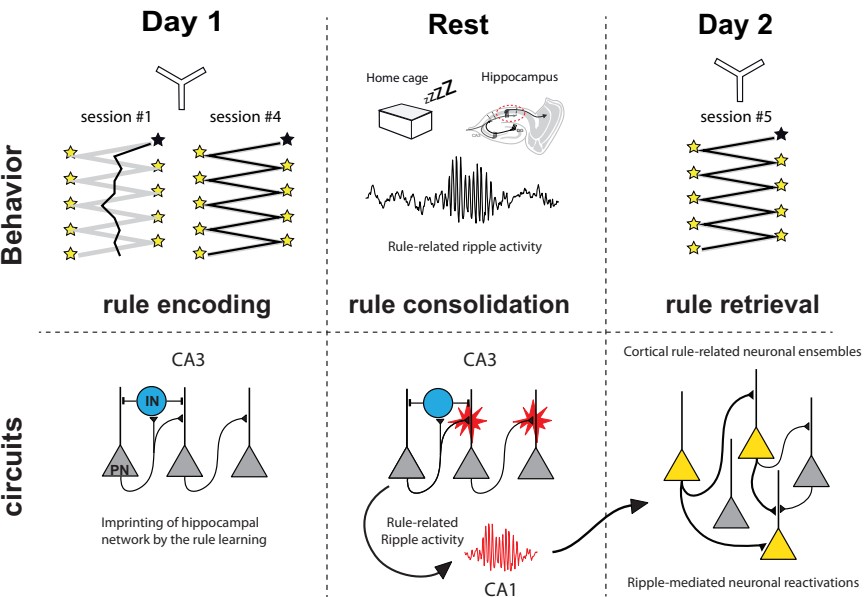

**Fig. 8 | Working model.** Model proposed for the action of AMPAR X-linking onto DSA memory consolidation. During SWS, ripples necessary for stabilization of rule-related ensembles in cortical areas (through replays-dependent reactivation of neuronal ensembles) would be impaired as plasticity at CA3→CA3 recurrent is impaired.

recordings (pre-learning injections). We also designed a number of control conditions that would account for this effect: we injected animals with either saline, various monovalent or divalent antibodies, but also with IgG without submitting the animals to DSA learning (Fig. 3). In all of these conditions, no changes in ripples properties were observed.

## Comparison between in vivo and in situ recordings

Another issue concerns the comparison of SWRs recorded in vivo and in vitro[6]. Indeed, in our in vivo recordings, we essentially characterized changes in frequency and amplitude of oscillations recorded in the CA1 region after a 150–250-Hz band-pass filtering (Figs. 3 and 7), whereas we focused on the occurrence frequency of CA3/CA1 sharp waves in situ (SPW-Rs, Figs. 4 and 5). Sharp waves are proposed to reflect the dendritic depolarization evoked by the synchronous activity of subgroups of excitatory afferents from the CA3 region, whereas the ripple oscillations are thought to be generated in the CA1 region, in response to the sharp wave-associated excitatory inputs[12]. In the retained in situ recordings, the vast majority of SPW-Rs are co-detected in CA3 and CA1 regions with a significant and stable delay (Fig. 4). In CA1 region, when applying a 150–250-Hz band-pass filtering to SPW-Rs, it eventually leads to the emergence of a ripple-like oscillation (Supplementary Fig. 3a) that however appears to be highly unstable. Another issue is that in some CA3 *stratum pyramidale* recordings, 150–250 Hz ripples were contaminated by recorded spikes (Supplementary Fig. 3c). Thus, the use of 150–250 Hz band-pass filtered ripples in situ appears to be less reliable than the associated co-detected waves. In vivo, the presence of dendritic responses—waves—is present in some but not all recordings, as the wires implanted are separated by 200 μm in the dorso-ventral axis, thus radially to the CA1 region (Supplementary Fig. 2a). However, because waves and ripples reflect intrinsically coupled network events, we believe that effects observed on in vivo ripples and in situ sharp wave frequencies are comparable. Notably, we have confirmed the fact that they display a similar lack of sensitivity to AMPAR X-linking in basal condition, whereas being impaired by the same treatment after DSA learning (in vivo, Figs. 3 and 7) or LTP induction (in situ, Fig. 5).

## Synaptic plasticity in memory phases

Surprisingly, the effect of AMPAR X-linking on ripple activity is present only after DSA rule encoding (Fig. 3). This suggests that ripple activity is different if salient cognitive events have to be consolidated. One intriguing possibility would be that the learning process directly or indirectly "conditions" or "imprints" the hippocampal network, especially among CA3→CA3 recurrent synapses, to drive DSA-associated ripples to be efficiently generated offline (Fig. 8). A likely mechanism by which this could happen is the occurrence of synaptic "tagging"—activity-dependent synaptic plasticity events—during online rule acquisition, that will have to be "captured" during the offline consolidation phase of memory[4,5]. It has yet been suggested that in vivo synaptic modifications would protect these synaptic contacts from ripple-mediated synaptic downscaling during sleep, thereby allowing cognitive map refinement[11]. Therefore, several aspects of our findings need to be discussed regarding this conceptual line, especially the fact that memory encoding seems to occur even if synaptic plasticity is blocked during rule encoding.

Synaptic tagging is thought to rely on cellular and molecular mechanisms associated with induction and expression of synaptic plasticity, which involves AMPAR trafficking at the plasma membrane[1]. However, we have already reported that AMPAR cross-linking was not impacting LTP induction[15] as NMDA receptors are kept functional. Thus, coincident neuronal activations may have activated transduction cascades necessary for synaptic tagging even if no LTP was expressed.

It remains quite surprising that we did not observe an impact of hippocampal AMPAR cross-linking on DSA rule encoding (Fig. 1d, left). It would suggest that synaptic tagging associated with rule acquisition does not depend on activity-dependent changes in synaptic strength in the hippocampus, whereas it is assumed to integrate all the components of an experience into one unique episodic memory[26].

One possible hypothesis is that the rule encoding may occur independently of AMPARM-dependent Hebbian plasticity. Some reports yet indicated that spatial map reorganization upon rule acquisition can be independent of NMDA-dependent plasticity. For example, Dupret et al.[27] injected rat hippocampus with an NMDAR antagonist in order to interfere with spatial memory. They observed that learning performance was unaffected but that animals failed to remember the newly learned locations. Their results suggested that the recently acquired representations of goal locations replayed during sleep did not stabilize[19]. Another recently described form of "non-Hebbian" plasticity that could contribute to DSA learning is behavioral

time scale synaptic plasticity that underlies CA1 place fields[28]. It is dependent on L-type VGCCs and NMDARs, and allows a broad−second scale−temporal relationship between pre- and post-synaptic activities[28]. However, it remains to be determined if this form of CA3→CA1 plasticity depends on the surface mobility of GluA2-containing AMPARs.

A second explanation would be that our cross-linking strategy did not perturb some hippocampal synaptic contacts crucial for rule encoding. Considering that our two strategies target GluA2-containing AMPAR. Therefore, a number of excitatory synapses may have escaped from the effect of cross-linking, such as excitatory inputs onto interneurons that can be GluA2 independent[29]. It is of note because some pyramidal cell-interneuron coupling changes have been reported during spatial rule encoding[30]. Thus, to have a complete and comprehensive view of AMPARM impact onto learning-dependent ripple physiology, further experiments using cross-linking strategies targeting GluA1-containing AMPAR, or specific cell types using conditional expression of the BiRA under IN or PN promoters will be necessary. Along the same line, we observed that DG→CA3 LTP was preserved in the presence of anti-GluA2 IgG, which would possibly allow rule encoding through large synaptic tagging within the CA3 region. Further in vivo experiments using specific blockers of DG→CA3 plasticity, such as Rp-cAMP (Supplementary Fig. 4) would allow deciphering the role of these particular connections in the encoding of the DSA rule in the hippocampus.

An alternative explanation for this unexpected dissociation between learning ability and blockade of LTP in the dHPC could reside in the resilience of the system. A recent study using another strategy to block post-synaptic LTP showed that even if CA1 plasticity was largely absent, and some of the learning-induced cellular rearrangements were lost, animals were still able to perform correctly[2].

Also, DSA rule can first be encoded in another brain region rather than the dHPC, such as being hosted in the medial prefrontal cortex (mPFC). For example, Peyrache et al. showed that in very similar conditions of a new rule learning in a Y-maze, ripple activity was directed toward the reactivation of rule-related neuronal ensembles in the mPFC[8], opening the possibility that synaptic tagging may have been generated there. Indeed, replay of firing patterns in hippocampal neuron ensembles during sleep is thought to cause the gradual formation of stable representations in extra-hippocampal networks by enhancing connectivity between their elements[26]. Thus, we can anticipate that pre-learning AMPAR X-linking experiments in the mPFC, by interfering with LTP-dependent synaptic tagging during encoding, would impair animal performance. Similar results would be obtained at pre-rest injections, by blocking ripple-mediated generation of DSA rule representations[8].

Interestingly, by using pharmacological strategies inactivating functionally calcium-permeant AMPAR (CP-AMPAR), Torquatto et al. showed that the presence of GluA1 homomers at memory-related synapses in the hippocampus is of crucial importance to mediate fast memory reactivation during memory recall/retrieval[31]. However, our results described here showed that GluA2-targeting AMPARM blockade did not deteriorate animal performance during retrieval of the DSA task 24 h after encoding. One would anticipate that a strategy targeting GluA1-containing AMPARM would possibly affect DSA memory retrieval. Our results thus add on those reviewed by Pereyra and Medina[32] suggesting that memory retrieval is a fast process probably because CP-AMPARs are present at potentiated synapses. The dynamic equilibrium of calcium-permeant to calcium-impermeant AMPARs at behaviorally relevant synapses supporting retrieval may depend on the type of memory, the structure and the memory consolidation state[32].

### Synaptic plasticity and memory maintenance
One interesting observation emerging from our data is the fact that in the absence of proper consolidation, the DSA rule is apparently completely forgotten, as if the animal would be completely naive to the task (Fig. 2). So far, experiments impacting ripples activity during sleep impaired the performance of the animal on the following days, slowing down the rising of behavioral performance[30], but were not reported to lead to complete resetting. An intriguing possibility would be that in the absence of synaptic capture, synaptic tagging would gradually fade away, the neuronal network returning to a "naive" state, as yet suggested by Norimoto et al.[11]. Interestingly enough, this would open a time window during which newly encoded memory would be accessible to be erased, in a state vulnerable to ripple physiology impairment.

This study is bringing the first piece of evidence that consolidation of recently acquired memory depends on AMPAR mobility in the hippocampus, especially pointing to the importance of the CA3 region in this process. Our results are embedded in the more global framework of the synaptic tagging and capture hypothesis that is now more and more discussed in terms of encoding/consolidation of memory during awake/sleeping state of the animals. The importance of the cortico-hippocampal reciprocal dialog in this process is of fundamental interest. Intriguingly, deciphering the intimate mechanisms of this dialog will certainly profit from the development and use of in vivo applicable molecular strategies interfering with plasticity in vivo with good temporal and spatial control.

## Methods
All procedures were validated by the ethical committee of animal experimental of Bordeaux Universities and the French Ministry of Agriculture (CE50; Animal Facility PIV-EXP, APAFIS#18507-201901118522837; Animal Facility A1, APAFIS#4552 2016031019009163; Animal Facilities Neurocentre Magendie and PIV-EXPE, APAFIS#13515- 2018021314415739).

### Biological models
Experiments in this manuscript were conducted on 6- to 12-week-old male mice belonging to two strains: C57BL6/J wild-type and C57BL6/J transgenic AP-GluA2 knock-In (KI, maintained on a C57BL6/J background) strains. Mice were kept on a 12-h light/dark cycle and provided with *ad libitum* food and water, except for food restriction associated with behavioral testing (see below). Mice were housed with 3−5 littermates except when demanded by the protocol. Temperature was between 20 and 24 °C and hygrometry was between 30 and 70%. The experimental design and all procedures were in accordance with the European Guide for the care and use of laboratory animals.

AP-GluA2 KI strain was developed and validated as a mouse model for AMPA receptors mobility in[16]. AP-GluA2 KI mice are similar to wild-type C57BL6/J mice in terms of weight, size, growth or fertility, but also for tested cognitive abilities[16]. At the genetic level, this strain presents a substitution of the endogenous GluA2 subunit of the AMPA receptor by a genetically modified one bearing an AP- (acceptor peptide) tag on the extracellular domain of the subunit. In the presence of the BirA ligase enzyme, which is not endogenously expressed, AP can bind Biotin, yet present in the murine brain. Thus, expression of AMPA receptors bearing biotinylated GluA2 subunits is restricted to neurons in which BirA ligase has been introduced by viral transfection, allowing targeting of AMPAR cross-linking. Indeed, the presence of extracellular tetrameric Neutravidin consecutive to intracranial administration leads to immobilization of AMPA receptors at the synaptic and perisynaptic space (see ref. 16).

### Surgery
Various surgery protocols were performed depending on the aim of the procedure, dividing into two major subgroups: stereotaxic injections and chronic implantations. They eventually shared some common steps, hereby listed. Surgery protocols were similarly applied for both mouse strains.

Mice were anaesthetized through exposure to the anesthetic gas agent, Isoflurane (4% mixed with air) for 4 min and anesthesia was maintained all throughout the surgery at 2% mixed with air. Mice were positioned in the stereotaxic apparatus (David Kopf Instruments) on a heating pad and received a subcutaneous injection of Buprenorphine (100 μL, 0.1 mg/kg) and a local injection of Lidocaine (100 μL, 0.4 mg/kg) for analgesia. The scalp was rinsed with Betadine to prevent infections. After incision and opening of the scalp, Bregma and Lambda point were identified in order to identify the region of interest using atlas coordinates (Paxinos). Finally, sutures were applied to close the incision point, and mice were subcutaneously injected with analgesic agent (Carprofen, 100 μL, 4 mg/kg), fed with powdered-nutrient-enriched food and left recovering inside a recovery cage positioned on a heating pad for 30–120 min. A post-surgery care routine was observed for 2–6 days following the surgery, during which the weight and general shape of mice were monitored and analgesic drugs were administered if needed.

Stereotaxic injection was performed on mice aged 6–10 weeks. Injected viral vectors were charged inside μL graduated glass Hirschmann pipets (ref. 960 01 05, Germany) and pressurized via a 5 mL syringe (Terumo). The pipette was automatically descended into the target region at a speed of approximately 20 μm/s (3 injection sites in the dorsal CA3; coordinates: AP −2.35; ML ±2.65; DV −2.5/−2.0/−1.5). Injection of 250 nL of the product was performed manually by applying low but constant pressure on the syringe. The pipette was maintained in position within the target region for 5 min after the end of the injection to allow local diffusion of the injected product, then retracted at a slow speed. When combined with other surgical procedures, stereotaxic injection always preceded stereotaxic implantation.

Chronic implantation of guide cannulas and/or electrodes was performed on mice aged 6–10 weeks. Various types of implants were used, all of which are detailed in the dedicated "Implanted materials" section, but surgical procedures were common to all. Prior to proper implantation, the skull was prepared by briefly applying Peroxidase RED ACTIVATOR (Super-Bond, Sun Medical Co) for 3-5 s to remove the periosteum. Single guide cannulas were manually descended into the target region at a speed of approximately 20 μm/s (coordinates CA1: AP −1.95; ML ±2.25; DV −0.55; angle 30°; coordinates CA3: AP −2.35; ML ±2.65; DV −1.2). Electrodes were descended into the target region using a micromanipulator at 1 μm/s for the last third of the descent, to reduce tissue damage caused by the implantation (coordinates: AP −2.35; ML ±2.3; DV −1.7; glued guide cannulas coordinates: AP −2.35; ML ±2.65; DV −1.2). Guide cannulas and electrodes were fixed with dental cement (Super-Bond, Sun Medical Co). After implantation mice were housed alone to prevent implants' damaging.

## Implanted materials

We used stainless steel guide cannulas (Bilaney 26 gauge, 1.5 mm of length; PlasticOnes). Prior to the surgery, guide cannulas were kept in alcohol to minimize the risk for bacterial contamination and plugs were maintained on them at all time to avoid penetration of external material. When intracranial injections had to be combined with extracellular field recording, guides were glued directly on the electrode connector and were obstructed with a metallic dummy cannula to avoid penetration of external material. Intracranial injection was performed on awake mice, either loosely held in the manipulator hands (for short injections) or free to move inside their home cage. Injections were performed through injection cannulas (Internal Cannula FIS 2.5 mm guide, Bilaney; 0.5 mm projection) and via an automatic pump (Legato 101, Kd Scientific Inc.) that applied a constant pressure using 1 μL Hamilton syringes (7101 KH), allowing the regulation of injection speed (antibodies: 100 nl/min; Neutravidin: 50 nl/min). Pre-learning and pre-test injections were performed 1 h before the beginning of the behavioral protocol; pre-rest injections were performed immediately after the last session of the behavioral protocol.

For hippocampal ripples recordings, bundles of Nichrome wires (diameter: 13 μm, Sandvik Kantal) were connected to an 18 male connector (nano 18 positions 2 guides ISC-DISTREL SA Omnetics) and were passed through a guide cannula (see Supplementary Fig. 2) to protect them from damage and spreading while entering into the brain.

## Chemicals

All viral vectors used for the experiments described in the Results section are Adenoviruses and their engineering is detailed in ref. 16. Ongoing production was assured either by the viral core facility of the Bordeaux Neurocampus IMN or by Charité Universitats medzin Berlin or viral vectors were ordered on Addgene. All viruses were stocked at −80 °C for long-term storage, conserved at 4 °C during surgery preparation and injected at room temperature. 250 nL per injection site of viral vector solution was administered through stereotaxic injection during surgery.

- pAAV9a-pSyn-BirA-ER-IRES-eGFP ($5.6 \times 10^{13}$ gcp/mL, IMN). The pSyn promoter allows the expression of the BirA enzyme in all neuronal types without distinction. BirA ligase expression promotes biotinylation of the extracellular portion of the GluA2 subunit of AMPA receptors, thus inducing AMPA receptors cross-linking in the presence of Neutravidin. eGFP is used as a tag to identify neurons expressing the enzyme.
- pAAV9a-pSyn-IRES-eGFP ($1.8 \times 10^{13}$ gcp/mL, IMN). Lack of BirA ligase coding sequence makes this viral vector a control for uncatalyzed Biotin binding to GluA2 subunits bearing the AP.

Production and conservation of antibodies used for the experiments detailed in the Results section of this manuscript is described in ref. 15. All antibodies were stored at −80 °C for long-term storage, conserved at 4 °C for a maximum of 1 week preceding injection and injected at room temperature. 500 nL per injection site of antibody solution was administered via intracranial injection in the awake, freely moving mouse.

- Antibody against GluA2 subunit of AMPA receptors (clone: 15F1; 2.9 mg/mL). This antibody is a monoclonal divalent IgG-κ directed against the extracellular domain of the GluA2 subunit of AMPA receptors. The divalent nature of this antibody allows for the binding of two target GluA2 subunits at the same time, therefore promoting AMPA receptors cross-linking. In vitro, a washout time of 8 h due to internalization of clustered receptors has been observed.
- Fragment Antigen-Binding (Fab; 2.9 mg/mL). The antigen-binding portion of the antibody directed against GluA2 subunits was isolated and used as monovalent control for cross-linking.
- Antibody against GFP (2.9 mg/mL). This antibody is a divalent IgG-κ from murine clones 7.1 and 13.1 (11814450001, Roche). As murine neurons do not physiologically synthetize GFP, this antibody was used as control for unspecific antibody binding.

Neutravidin or NA: Texas Red-conjugated Neutravidin (8.33 μM; Invitrogen, A2665) was used to operate cross-linking of AMPA receptors in the AP-GluA2 KI mouse model. 500 nL per injection site of Neutravidin solution was administered via intracranial injection in the awake, freely moving mouse.

Monomeric streptavidin or mSA was produced and conjugated to STAR 635P (Abberior, ST635P) using N-hydroxysuccinimide ester–activated fluorophore coupling as previously described. 500 nL per injection site of mSA solution (concentration: 8.33 μM) was administered via intracranial injection in the awake, freely moving mouse.

Saline physiological solution: saline physiological solution was used as a control for cross-linking in the AP-GluA2 KI mouse model. 500 nL per injection site of saline solution was administered via intracranial injection in the awake, freely moving mouse.

## Behavioral protocol

Delayed spatial alternation (DSA) task: the DSA task is a delayed non-matching-to-place task used to assess special navigation and cognitive functions in rodents[17].

Food restriction was required to ensure mice's motivation. Mice were weighted right before food withdrawal and this weight was used to calculate the 85% of weight-loss limit that was fixed for protocol termination. On the first day of restriction, mice were fed with Perles pasta of the same type as those that were used to bait the maze during the behavioral task, in order to habituate them to the new food. On subsequent days, mice were fed at the end of all behavioral manipulation with 2–3 g of powdered nutrients-enriched food, in order to maintain them to about 85% of their initial weight.

A custom-made, semi-transparent white PVC Y-maze was used for the task. All three arms are identical (40 cm length, 8 cm width, 15 cm high walls), except for an additional closable rectangular chamber (15 cm × 25 cm) bridged to the "Starting arm". Arms are spaced by a 120° angle. An opaque small container was positioned at the end of each "Goal arm" to serve as food well for reward delivery. Environmental cues are positioned on the room walls surrounding the maze. Video recordings are realized through an infrared camera (Basler USB camera – ac1920-155um – Noldus) positioned on the ceiling upon the center of the maze. The DSA tests were realized in conditions of dim light.

Habituation lasted for 5–8 days, depending on each individual mouse, and was divided into 3 phases. The first phase, starting before food restriction, consisted of 2–3 days of handling in order to habituate the mouse to be manipulated, especially for injection cannulas insertion and/or electrodes plug-in. Proper habituation for the task started on the second day of food restriction and consisted of multiple sessions of free exploration of the maze. The sessions were stopped when the mouse had eaten food-pellet in all three baited arms. The last phase of habituation consisted of a single trial in which the mouse was positioned inside the "starting arm" (defined by position with respect to environmental visual cues) and had to collect a reward food-pellet in each of the two "goal arms", with a time-limit of 1 min.

The DSA task consists of 10 trials in which the left and right goal arms are alternatively baited with a rewarding pellet. During the first trial, the choice is forced toward the baited arm, setting the pattern of alternation (i.e., the reward zone of the un-baited arm is made inaccessible through positioning a PVC slide at the entrance of the proximal portion of the arm; each consecutive session alternatively starts with a forced right or left choice). The nine following trials rely on the mouse-free choice of one of the two arms. A single trial can be repeated up to 5 times ("runs") if the mouse makes consecutive mistakes, of which the sixth consists in a forced run in the baited arm direction. Once the mouse has reached the reward zone of the chosen arm, access to the reward zone of the unchosen one is restricted and the mouse is let spontaneously come back to the distal portion of the starting arm. After every run, the mouse is placed back in his home cage and a delay of 30 s is respected before the mouse is allowed to explore the maze again. During this delay period, the maze is cleaned with ethanol to prevent odor-based navigation. On the first day of training, 4 sessions are conducted, spaced by 30–60 min rest period; on the 2 following days, 2 sessions per day are realized, spaced by 30 min (see Fig. 1). Behavioral training was conducted between 8 a.m. and 1 p.m.

Graphs with random choice (dashed line) at 1: in order to get a "trial-based" resolution, we choose to evaluate the error rate of each trial position. We thus cumulated the number of errors—a maximum of 5 error per runs is authorized—that was done by all animals at a given trial position—session #1 trial #4, for example—and divided it by the number of animals. Thus, if 23 errors were cumulated at session #1 trial 4 by 17 animals, the error rate is 23/17 = 1.35. The error bars are the variability of this error rate among the run of the session (trial #2, #3, #4, etc.) and expressed as SEM. For graph with random choice (dashed line) at 0.5: the calculation is Errors/total for an entire session, and the variability is the average of all mice performance for this session (±SEM).

Vicarious trial and errors (VTE) and non-VTE run sorting were done by double-blinded visual detection as in ref. 17. Cross-controlled visual detection of the wavering behaviors of the mouse at the center of the maze was done on videos by independent experimenters. Recurrently detected VTE trials were kept for final selection. Comparison of the manual sorting to the measurement of instantaneous angular speed as in ref. 18 showed a highly significant difference in ldPHI and zldPhi scores between our sorted VTE and non-VTE trials (Mann–Whitney Rank sum test, $p < 0.001$). However, because this measure was not sensitive enough for our behavioral conditions and video recordings, we kept our manual, double-blinded video analysis.

## In vivo electrophysiological recording

Electrophysiological recordings are realized by plugging a headstage (INTAN) containing 16 unity-gain operational amplifiers into each connector. Recordings were realized through the recording system OpenEphys. Recordings were performed during resting periods in the mouse home cage positioned in an isolated closed box, allowing cable suspension and infrared recordings by the camera (Basler USB camera – ac1920-155um – Noldus). A baseline is recorded for 3 h during a rest session the day preceding the first training sessions. This recording is repeated the following day after the 4 sessions of training. They start on average 10 min after the end of the last habituation/training session of the day.

Electrophysiological data were imported on Matlab and down-sampled to 1 kHz for storage and analysis speed convenience. Ripples detection was performed on Matlab scripts originally developed by Cyril Dejean. At first, referencing and band-pass filtering at 50 Hz eliminates noise oscillations common to all channels. Then, 100–250 Hz band-pass filtering is used to detect ripple events that are selected if respecting the following criteria: (1) its amplitude is higher than 5 standard deviations of the mean band-passed trace, (2) the event must be at least 30 ms long and (3) two ripples must be separated by an interval of at least 45 ms. Ripple's characteristics are then computed, including timestamp of the peak, intrinsic frequency, number of oscillations, mean amplitude (both on the filtered and the integrated trace), area under the integrated curve, duration (total and of each part preceding and following the peak), half prominence.

## In situ slice recordings

Male C57bl6/J mice were used at the age of 4–9 weeks. The extracellular artificial cerebro-spinal fluid (ACSF) solution utilized for slice recordings is composed of: 119 mM NaCl; 2.5 mM KCl; 1.3 mM $MgCl_2$; 2.5 mM $CaCl_2$; 10 mM glucose; 1 mM $NaH_2PO_4$; 26 mM $NaHCO_3$. The cutting solution is an ice-cold sucrose solution (1–4 °C) composed of 2 mM KCl; 2.6 mM $NaHCO_3$; 1.15 mM $NaH_2PO_4$; 10 mM glucose; 120 mM sucrose; 0.2 mM $CaCl_2$ and 6 mM $MgCl_2$. Both solutions are oxygenated with carbogen (95% $O_2$, 5% $CO_2$, pH 7.4 at 37 °C, 290–310 mOsm/L). For brain dissection, mice are anesthetized with 5% isoflurane for 2 min before decapitation. The head is immersed in the iced sucrose solution. The removed brain is immersed for 4 min in iced oxygenated sucrose solution and then placed on a cellulose nitrate membrane to separate and position the hemispheres in the vibratome (Leica VT1200s) to obtain 400 μm horizontal slices (cutting speed of 0.1 mm/s). Once produced, slices are semi-immersed in a dedicated incubation chamber, oxygenated and maintained at 35 °C for at least 2 h before starting the recordings.

Recordings are made in an S-shaped recording chamber, maximizing oxygenation while preventing slice movement caused by the

3.5 mL/min perfusion flow. Field recordings are obtained using glass micropipettes stretched with a PC-10 (Narishige, Japan) and broken at their tip to decrease the resistance (<0.5 MΩ). Depending on the experimental configuration, the pipette is filled either with ACSF or supplemented with antibodies, IgG α-GluA2, 15F1 (IgG) or IgG Fab (Fab). Electrophysiological recordings are obtained by a MultiClamp 700B (Molecular Devices, Foster City, CA) using Clampfit software (Molecular Devices, Foster City, CA). Electrical stimulation is provided by a CBCSE75 concentric bipolar electrode (FHC, Phymep, France) and an external A.M.P.I Iso-flex stimulator. Synaptic fields are recorded in the *stratum radiata* (sr) of CA1 and CA3 to measure evoked fEPSPs, and to test the presence of propagated SPW-Rs. Depending on the configuration, stimulation electrodes are placed in the *stratum radiatum* of CA1 or CA3 to stimulate respectively CA3→CA1 and CA3→CA3 axons and elicit basal (0.1 Hz) or high-frequency stimulations (HFS, 100 Hz, 1-s train repeated 3 times each 30 s) to induce long-term potentiation (LTP). When stimulating CA3-sr, a train of 10 stimulations at 1 Hz is first applied to test for eventual contamination by dentate gyrus mossy fibers that display frequency facilitation.

## Perfusion and histology

Mice were anaesthetized with a mix of Ketamine and Xylazine (100 mg/20 mg/kg) diluted in a saline solution; 10 μL of solution per gram weighted by the animal were administered via peritoneal injection. Perfusion with paraformaldehyde (PFA, concentration: 4%) was realized on the anaesthetized mouse and the brain was initially collected without being evicted from the scalp. After 48 h of storage in PFA at 4 °C temperature, implants and scalp were removed and the brain was washed three times in PBS solution (1%) and then stored in PBS for 24–72 h at 4 °C. Slicing was performed with a vibratome (Leica VT1200s). Coronal slices of 60 μm thickness were collected at a speed of 30–50 mm/s from the regions of interest and stored in PBS for 24 h before being mounted on slides and covered with Fluoromont-G (complemented with DAPI for cellular nuclei staining, Thermo Fisher Scientific).

Image acquisition of slides was performed with an epifluorescence microscope (Nikon Eclipse NI-U) coupled to an illumination system (Intensilight C-HGFI, Nikon) and a camera (Zyla sCMOS, Andor Technology, Oxford Instruments).

## Reporting summary

Further information on research design is available in the Nature Portfolio Reporting Summary linked to this article.

## Data availability

All single data and scripts used for analysis can be found at the following address: https://figshare.com/account/home#/projects/179595. The relevant raw data from each figure or table (in the main manuscript and in the Supplementary Information) are represented by a single sheet in an Excel document. All reagents are available upon request. The mouse line can be provided pending a completed material transfer agreement. Requests for all materials should be submitted to Y.H. at yann.humeau@u-bordeaux.fr. Source Data are provided with this paper.

## Code availability

The scripts that have been used for ripple analysis can be found at the following address: https://figshare.com/account/home#/projects/179595. For further information, please contact the corresponding author.

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

## Acknowledgements

We thank T.-A. Vernoy, G. Dabee, E. Normand, P. Costet, M. Dehors, C. Martin, for support with animal husbandry and in vivo experiments; S. Marais for imaging and analysis support. This work was supported by European Research Council (ERC) grants to D.C. (grants ADOS 339541 and Dyn-Syn-Mem 787340), a Fondation Recherche Médicale (FRM) grant to Y.H. (grant DEQ20180339189 AMPA-MO-CO), and Agence Nationale de la Recherche (ANR) grant to Y.H. (grant OptoXL ANR-16-CE16-0026; EXINmemory ANR-22-CE37-0025; INEXCONSO ANR-21-CE37-0034). This work was supported by the Bordeaux Neurocampus core facilities (LabEx BRAIN; grant ANR-10-LABX-43), including the In Vivo Experimental (PIV-EXPE) facilities of the IINS (CNRS); the biochemistry and biophysics platform and animal genotyping facility of the Neurocentre Magendie (INSERM). The microscopy was done at the Bordeaux Imaging Centre, a service unit of CNRS-INSERM and Bordeaux University, a member of the national infrastructure France BioImaging supported by the French National Research Agency (grant ANR-10-INBS-04).

## Author contributions

Experimental conception and design: D.C., C.H. and Y.H. Manuscript preparation: Y.H. All authors discussed the results and edited the manuscript. Behavioral experiments: H.E.O., C-L.Z., U.F., C.C. and A.L.-S.-A. Electrophysiology: U.F., F.L., P.M., L.V. and Y.H. Animal surgeries: H.E.O., C-L.Z., A.L.-S.-A., U.F., and A.N. Data analysis: M.L. and C.D.

## Competing interests

The authors declare no competing interests.
