## [Peer Review File · Nature Communications]

CA3 hippocampal synaptic plasticity supports ripple physiology during memory consolidationREVIEWER COMMENTS

Reviewer #1 (Remarks to the Author):

This paper designed a series of in vivo and in situ experiments utilizing the cell-surface AMPA receptor immobilization technique to test the role of cell-surface AMPAR in memory formation, consolidation and retrieval in the dorsal hippocampus. The Authors found that 1. Blocking cell surface AMPAR will impair memory consolidation but not memory formation or recall in their delayed spatial alteration task (DSA). 2. In situ experiment showed the AMPAR immobility induced impairment of activity (ripples) and plasticity happened specifically at CA3-CA3 recurrent synapse. 3. Blocking AMPAR mobility in CA3 only reproduced the upper-mentioned behavior result, indicating AMPAR mobility at CA3 recurrent synapse support the generation of ripples and memory consolidation. The work is in general well conducted and provides new evidence of AMPAR's critical role, however, I have concerns about the novelty of the finding, the types of synaptic plasticity discussed in this paper and the lack of description of all the statistics that have been used in this study, which I have detailed below.

1. I have some concerns regarding novelty, as the important role of AMPAR in synaptic plasticity and memory is well acknowledged in the field. Specifically, for the AMPAR's role in memory consolidation has also been investigated, such as Alvares et.al 2019. Maybe the author could further state the role of cell-surface AMPAR in this process.

2. With more and more understanding of the hippocampus, the field has acknowledged that the synaptic plasticity that happens in the hippocampus could be Hebbian and non-Hebbian, and different types of plasticity have different molecular mechanisms. Would the author suggest the plasticity that they are testing here is more related to classic Hebbian plasticity (based on the analysis in the in-situ result)? I would suggest the authors provide readers with a better definition of what type of plasticity they are referring to and why.

3. For the experiment that is described in figure1, 'pre-learning injection' and 'pre-rest injection' seem to induce similar results, could their authors share their interpretation from an experimental point of view? Is it possible that the drug effect is not acute and the bad performance on day2 is a mix of impairment of memory consolidation and recall? The reviewer assumes the authors were comparing the IgG and FaB group but also notice that for the 'post-test injection' group session 6 performance is worse than session 5, did the author try to test if it's significantly different?

4. One relatively big problem that make a lot of the analysis that is performed in this paper hard to review is that the authors didn't clarify what kind of statistical tests were used in each individual analysis. For example, in comment#3, the reviewer can't tell what kind of statistical test is performed and it was hard to interpret the result. This should be addressed clearly.

5. The authors showed blocking the cell-surface AMPAR mobility only influences memory consolidation, however, there are also a lot of studies has shown the critical rule of AMPAR in memory retrieval (see review Medina 2021), I wonder if the authors could provide their thoughts on previous result and their finding, maybe from the angle of different task design that focusing on different features of memory, for example reward coding vs fear memory, new rule formation vs rule updating etc.

Other minor points:

Figure 1d, Could the author clarify why the mean error rate (number per trail) in the left panel is higher than 1? Also, what is the error bar?

Figure 2, Could the author clarify the standard for VTE and non-VTE run in method?

Figure 2d figure inset seems to be wrong, should the red line be the IgG group?

The resolution of the figures is relatively low, specially in figure3.

Line 175 to 192, unless this is a new technique to the field, it seems more to belong to the method part, also I suggest figure4 a-c be moved to supplementary figure if it's not directly related to the result. This might be a personal preference, so feel free to ignore it.

Figure 5b bottom, what is a,b,c?

Figure 5c is a bit hard to follow, especially the bottom figure, what's the relationship between SWRs frequency and fEPSP amplitude? Although the author mentioned co-evolution, I wonder if it will make the readers confused about the relationship between SWR amplitude and fEPSP frequency as the percentage converged after 10 minutes, which as I understand, is just a coincidence?

Figure 6, I don't think it's a requested experiment to do, but I'm wondering if the authors have tried a similar experiment in CA1 and if it won't produce the same effect as in CA3.

Sup Figure1:

Is red and blue indicate FaB and IgG here as well? Also, figure legend has two 'session's, which seems to be a typo.

Sup Figure2:

The label of sub-figure is misleading, where is c? And I think d is the subfigure c? and ii and iii are e and f?

Reviewer #2 (Remarks to the Author):

This study presents data on how post-synaptic AMPAR mobility at CA3 recurrent synapses supporting the generation of ripples necessary for rule consolidation. When a rule has been encoded, a strong impact of AMPAR immobilization on ripples measured by in vivo recordings during resting periods. In situ examination of the interplay between AMPAR mobility, the results showed post-synaptic plasticity at CA3-CA3 recurrent synapses support ripple generation. The study makes an addition to a growing literature on hippocampal synaptic plasticity and memory consolidation.

Why did the authors investigate the role of AMPAR in synaptic plasticity of the memory consolidation, but not the NMDAR or any other receptors? The importance of AMPAR and the reason for this study are needed in the introduction.

Line 224, the authors made the conclusion "the effect of HFS on synaptic strength and SPW-Rs frequency seems to be temporally disconnected", because the fEPSP and ripples were not enhanced simultaneously after the HFS. However, the mechanism of ripple generation is complicated, which is not only modulated by the presynaptic high frequency stimulation. So this conclusion should be toned down, because whether they are "temporally disconnected" cannot not only be measured by the increment in the first 5 mins. In addition, the unit was lack for "0-5 post-tetanic period" in Line 225. Line 587, "Yellow zone: time after HFS application" cannot be found in the figure.

In Fig1c, a subfigure was missing according to the figure legend.

In Fig2, b-e showed the data repeatedly, which is not necessary. In the Fig2b, the marker of VTE runs and no VTE runs look very similar.

In Fig3a bottom, the y-label is not correct. It should be like ripple occurrence rate (Hz).

In the analysis of Fig3b-right and Fig 3e, because the ripple amplitude has been inhibited by IgG,

if the same criterial was used to detect ripples, the ripple frequency was certainly decreased. How to rule out the effect of amplitude on ripple detection? Also, the examples in Fig 3b-right was not consistent with this conclusion.

Line 227-228, again, this conclusion should be made more cautiously. The response of synaptic plasticity occurred earlier than that of the ripples, doesn't necessarily point to the causality between them.

Line 267, the results showed that the accuracy of VTE runs was improved in control mice. However, the error rate was higher in VTE5-6 than in VTE 1-2 in Fig 6e. Is there a mistake on the labels in the figure?

In the Fig 8, the synaptic mechanism in the cortex was included in the model, however, there was no findings related to the cortex in this study.

There is no description for statistics. The stats (hypothesis tests) in all figures should be shown. For example, what's the stats in fig 1D? There is no main effect or interaction for 2-way ANOVA, but only the multiple comparison was labeled in the figure. What is the sample size N in Fig 1g? It the N too small to do stats?

REVIEWER COMMENTS

Reviewer #1 (Remarks to the Author):

This paper designed a series of *in vivo* and *in situ* experiments utilizing the cell-surface AMPA receptor immobilization technique to test the role of cell-surface AMPAR in memory formation, consolidation and retrieval in the dorsal hippocampus. The Authors found that 1. Blocking cell surface AMPAR will impair memory consolidation but not memory formation or recall in their delayed spatial alteration task (DSA). 2. *In situ* experiment showed the AMPAR immobility induced impairment of activity (ripples) and plasticity happened specifically at CA3-CA3 recurrent synapse. 3. Blocking AMPAR mobility in CA3 only reproduced the upper-mentioned behavior result, indicating AMPAR mobility at CA3 recurrent synapse support the generation of ripples and memory consolidation. The work is in general well conducted and provides new evidence of AMPAR's critical role, however, I have concerns about the novelty of the finding, the types of synaptic plasticity discussed in this paper and the lack of description of all the statistics that have been used in this study, which I have detailed below.

1. I have some concerns regarding novelty, as the important role of AMPAR in synaptic plasticity and memory is well acknowledged in the field. Specifically, for the AMPAR's role in memory consolidation has also been investigated, such as Alvares *et al.* 2019. Maybe the author could further state the role of cell-surface AMPAR in this process.

AR: We would like to thank the referee for their positive opinion of our work, and his/her constructive comments that will improve the manuscript. Referee is right mentioning that numerous studies have examined the respective role of GluA-subunits in the learning process, and the one mentioned here, by Torquatto and Colleagues (10.1016/j.neuropharm.2018.10.030), is of great interest, asking the role of CP-AMPA in the consolidation and retrieval of various forms of recent memories according to their level of aversive content. Actually, we think that their results are in line with ours regarding memory retrieval. By using pharmacological strategies targeting CP-AMPA, they showed that the presence of GluA1 homomers at memory-related synapses in the BLA and hippocampus may be crucial to mediate fast memory reactivation during recall. From our work, we conclude that memory encoding and memory retrieval do not depend on GluA2-containing AMPAR mobility, and its related forms of plasticity. We also discussed a potential side effect of our antibody strategy in generating a switch between heteromeric GluA2-containing towards GluA1 homomers (in page 9 of the manuscript). Thus, it is still coherent with Alvares' lab findings.

Most importantly, we also want to point here that, in previous studies using these anti-GluA2 antibodies (Penn *et al.*, 2017) or GluA2-AP KI/BiRA strategy (Getz *et al.*, 2022), we never observed any effect on basic synaptic properties by blocking AMPAR mobility, and thus attributed the effects to the blockade of plasticity, rather than the synaptic function of the receptor itself, which is the case with classical pharmacological approaches such as the NASPM (CP-AMPA antagonist) used in Torquatto *et al.*

We add the following paragraph in the new version of the manuscript (page 13): Interestingly, by using pharmacological strategies inactivating functionally calcium-permeant AMPAR (CP-AMPA), Torquatto and colleagues showed that the presence of GluA1 homomers at memory-related synapses in the hippocampus is of crucial importance to mediate fast memory reactivation during memory recall/retrieval²⁶. However, our results described here showed that GluA2-targeting AMPAR blockade did not deteriorate animal performance during retrieval of the

DSA task 24 hours after encoding. One would anticipate that a strategy targeting GluA1-containing AMPARM would possibly affect DSA memory retrieval. Our results thus add on those reviewed by Pereyra and Medina²⁷ suggesting that memory retrieval is a fast process probably because CP-AMPARs are present at potentiated synapses. The dynamic equilibrium of calcium-permeant to calcium-impermeant AMPARs at behaviorally-relevant synapses supporting retrieval may depend on the type of memory, the structure and the memory consolidation state²⁷.

2. With more and more understanding of the hippocampus, the field has acknowledged that the synaptic plasticity that happens in the hippocampus could be Hebbian and non-Hebbian, and different types of plasticity have different molecular mechanisms. Would the author suggest the plasticity that they are testing here is more related to classic Hebbian plasticity (based on the analysis in the in-situ result)? I would suggest the authors provide readers with a better definition of what type of plasticity they are referring to and why.

AR: We fully agree with the referee that it is difficult to resume our findings to the sole blockade of “classical” pre/post Hebbian plasticity. We already mentioned in page 11 that “Our two strategies target GluA2-containing AMPAR. Therefore, a number of excitatory synapses may have escaped from the effect of cross-linking, such as excitatory inputs onto interneurons that can be GluA2 independent²⁷”, and later page 12 that “Along the same line, we observed that DG→CA3 LTP was preserved in presence of anti GluA2 IgG, that would possibly support rule encoding and relay broad synaptic tagging within the CA3 region.” Another recently described form of “non-Hebbian” plasticity that could support DSA learning is “behavioural time scale synaptic plasticity” that underlies CA1 place fields (Bittner et al, 2017). It is dependent on L-type VGCCs and NMDARs, but allows a broad – second scale - temporal relationship between pre- and post-synaptic activities. It is not yet known if this form of CA3→CA1 plasticity depends on surface mobility of GluA2-containing AMPARs. Currently, in the lab, we are exploring non-Hebbian forms of plasticity in hippocampal interneurons using IN-targeted KI strategies. Our results are in progress, and will be the core of a future manuscript.

We added a new paragraph to further discuss Hebbian VS non Hebbian plasticity in the page 12 of the new version. “Another recently described form of “non-Hebbian” plasticity that could contribute to DSA learning is behavioral time scale synaptic plasticity that underlies CA1 place fields³⁰. It is dependent on L-type VGCCs and NMDARs, and allows a broad – second scale - temporal relationship between pre- and post-synaptic activities³⁰. However, it remains to be determined if this form of CA3→CA1 plasticity depends on surface mobility of GluA2-containing AMPARs.”

3. For the experiment that is described in figure1, ‘pre-learning injection’ and ‘pre-rest injection’ seem to induce similar results, could their authors share their interpretation from an experimental point of view?

AR: Concerning the identical effect of pre-learning and pre-rest injections, we attribute it to the efficient blockade of AMPARM for several hours, that in both cases would include memory consolidation (see page 10: “In fact, the impact of both strategies blocking AMPARM appears to be specific on the actual cross-link capacity (see the various control conditions for both antibody- and neutravidin-based strategies) at the time of offline rule consolidation.”).

Is it possible that the drug effect is not acute and the bad performance on day2 is a mix of impairment of memory consolidation and recall? The reviewer assumes the authors were comparing the IgG and FaB group but also notice that for the ‘post-test injection’ group session 6 performance is worse than session 5, did the author try to test if it’s significantly different?

AR: We were actually very pleased to see that IgG procedure was not affecting DSA retrieval that would have complicated data interpretations. It suggests that there is no major impact of GluA2-containing AMPARs immobilization in memory retrieval, in good coherence with the described literature (see above discussion and new paragraph page 10). However, when we compare sessions #5 and #6 of pre-test IGG, there were indeed significantly different (Wilcoxon Signed Rank Test; $P = 0,016$). However, we attribute this difference to the exceptional performance of IGG-treated animals at session #5: Indeed, by including the 9 runs of session 5 for the 7 animals, only 8 errors were counted, making them the best performers among all groups, far above the performance of controls (Mann-Whitney Rank Sum Test; $P = 0,003$ as compared to all Fab controls).

Nevertheless, we agree with the referee that in case of a delayed effect of the drug, we could have observed an effect on the second but not the first retrieval test, leading to a better performance at session #5 as compared to session #6. However, multiple *in vitro* and *in vivo* evidences suggested us that the effect of antibodies on brain tissues is maximal within minutes, and thus would be fast enough to block retrieval at session #5, that is performed at least 60 minutes after drug injection (already mentioned in methods, page 30). In Penn *et al.*, 2017, *in situ* experiments using acute brain slices were performed with pressure-mediated injections of antibodies that were done 10-15 minutes before inducing plasticity, and we observed a complete blockade of the potentiation (Penn *et al.*, 2017). In this MS – Figure 4 and 5, supp Figure 4 – using the same methodology, we tested the effect of antibodies on SWRs and synaptic plasticity and observed significant effects within minutes. *In vivo*, the delivery of antibodies is done by large injection cannula, and is achieved in 5-10 minutes, covering a large proportion of the dorsal hippocampus (Figure 1). So, we believe that injected antibodies would lead to a fast and efficient blockade of AMPARM-dependent plasticity within 10-20 minutes *in vivo*.

We modified the paragraph in the new version of the MS (page 4): “To mediate AMPAR immobilization in the dorsal hippocampus, we performed bilateral, intra-cerebral injections of AMPAR cross-linkers (anti-GluA2 IgGs) or their controls (anti-GluA2 monovalent Fabs) at key times of the learning process, with a sufficient delay (> one hour) between IgG injections and behavioral testing to allow efficient AMPARM blockade (Figure 1b-c): immediately before the first learning session of day 1 (Pre-learning), immediately after the end of the first training day (Pre-rest), and immediately before the first session of day 2 (Pre-test). Our aim was to test the importance of hippocampal AMPARM-dependent plasticity in the encoding, the consolidation and the recall of DSA rule respectively. Collectively, our results pointed to an impact of AMPAR cross-linking onto memory consolidation. Indeed, pre-learning injections of AMPAR cross-linkers did not impact animal performance on day 1 (Pre-learning; Figure 1d left and 1e left), but rather on the following day, characterized by mice’s choices returning to random level (Figure 1d left and 1e right). A similar effect was observed when injections were performed immediately after session#4 (Pre-rest; Figure 1d middle, and 1f), but not if done before the test performed in day2 (Pre-test; Figure 1d right and 1g). Thus, the results indicated that memory retrieval was not impacted by AMPAR cross-linking, while pointing that AMPARM-dependent process occurs during resting period that is thought to support memory consolidation.”

4. One relatively big problem that make a lot of the analysis that is performed in this paper hard to review is that the authors didn’t clarify what kind of statistical tests were used in each individual analysis. For example, in comment#3, the reviewer can’t tell what kind of statistical test is performed and it was hard to interpret the result. This should be addressed clearly.

AR: We agree with the referees that our manuscript was missing a systematic explanation of the

statistical tests used. We now provide this information in the figure legends, and summarize them in a table provided as a supplementary data (supplementary table) added to the MS.

5. The authors showed blocking the cell-surface AMPAR mobility only influences memory consolidation, however, there are also a lot of studies has shown the critical rule of AMPAR in memory retrieval (see review Medina 2021), I wonder if the authors could provide their thoughts on previous result and their finding, maybe from the angle of different task design that focusing on different features of memory, for example reward coding vs fear memory, new rule formation vs rule updating etc.

AR: We thanks the referee to push forward this interesting discussion point. In this paper, we solely tested the effect of dorsal hippocampus GluA2-containing AMPARM in a non-aversive, spatial task, and found no significant effects on animal performance upon pre-test injections (Figure 1, injection done one hour before the first session#5 test). Beyond the discussion on the timing of the drug effect (see added sentence page 4), our results would then suggest that DSA retrieval would not depend on GluA2-containing AMPAR mobility at cell surface. With regards to other studies published on the role of AMPAR in memory retrieval and summarized by Peyreira and Medina, which principally targeted the function or modify the balance between AMPAR subunits at cell-surface by changing their cellular expression or trafficking, our work is based on acute depletion of the pool of receptors available to on-demand plasticity, without changing the basal levels (Penn et al., 2017; Getz et al., 2022). As pointed by Medina and colleague, memory retrieval is a fast process probably hold by CP-AMPARs that are present at potentiated synapses. Retrieval also opens a new window for plasticity, leading to eventual memory modifications, and the dynamic of CP to CI-AMPAR may depend on the type of memory, the structure and the memory consolidation state. It is an interesting topic, but we feel that our paper mostly focusses on the role of synaptic plasticity in ripple physiology which is heavily CA3-dependent – during rule consolidation.

We add the following paragraph in the new version of the manuscript (page 13): “Interestingly, by using pharmacological strategies inactivating functionally calcium-permeant AMPAR (CP-AMPAR), Torquatto and colleagues showed that presence of GluA1 homomers at memory-related synapses in the hippocampus is of crucial importance to mediate fast memory reactivation during memory recall/retrieval²⁶. However, our results described here showed that GluA2-targeting AMPARM blockade did not deteriorate animal performance during retrieval of the DSA task 24 hours after encoding. One would anticipate that a strategy targeting GluA1-containing AMPARM would possibly affect DSA memory retrieval. Our results thus add on those reviewed by Pereyra and Medina²⁷ suggesting that memory retrieval is a fast process probably because CP-AMPARs are present at potentiated synapses. The dynamic equilibrium of calcium-permeant to calcium-impermeant AMPARs at behaviorally-relevant synapses supporting retrieval may depend on the type of memory, the structure and the memory consolidation state²⁷”.

Other minor points:

Figure 1d, Could the author clarify why the mean error rate (number per trail) in the left panel is higher than 1? Also, what is the error bar?

AR: Graphs with random choice (dashed line) at 1: In order to get a “trial-based” resolution, we choose to evaluate the error rate of each trial position. We thus cumulated the number of errors (a maximum of 5 error runs is authorized) that were done by all animal at a given trial position - session#1 trial #4 for example – and divided it by the number of animals. Thus if 23

errors were cumulated at session#1 trial 4 by 17 animals, the error rate is $23/17=1,35$. The error bars are the variability of this error rate among the run of the session (trial #2, #3, #4, etc...) and express as SEM.

For graph in which the random choice is at 0,5: the calculation is Errors/total for an entire session, and the variability is the average of all mice performance for this session (+/- SEM).
This precision is now added in the methods section (page 32/33)

Figure 2, Could the author clarify the standard for VTE and non-VTE run in method?

AR: VTE/nonVTE trial sorting was done as for Zhang et al, 2017 (doi: 10.1523/JNEUROSCI.0351). It is based on cross-controlled visual detection of wavering behaviors of the mouse at the center of the maze with videos. Recurrently detected VTE trials were kept for final selection. We then compared our sorting with measurement of instantaneous angular speed as measured in: Redish, A. D. Vicarious trial and error. *Nat. Rev. Neurosci.* 17, 147–159 (2016), and indeed, 80% of VTE detected by the $zldPhi > 0$ were also VTE runs in our analysis (127/158, n=5 mice, 20 sessions). There was also a highly significant difference in $ldPHI$ and $zldPhi$ scores between our sorted VTE and non VTE trials (Mann-Whitney Rank sum test, $p < 0,001$). However, because this measure was not sensitive enough for our behavioral conditions and video recordings, providing a lot of false negative, we kept our manual, double blinded video analysis.

A paragraph has been added in the method section (page 33): Vicarious Trial and Errors (VTE) and non-VTE trial sorting was done by double blinded visual detection as in Zhang et al, 2017 (doi: 10.1523/JNEUROSCI.0351). Cross-controlled visual detection of wavering behaviors of the mouse at the center of the maze was done on videos by independent experimenters. Recurrently detected VTE trials were kept for final selection. Comparison of the manual sorting to measurement of instantaneous angular speed as in ¹⁸ showed a highly significant difference in $ldPHI$ and $zldPhi$ scores between our sorted VTE and non VTE trials (Mann-Whitney Rank sum test, $p < 0,001$). However, because this measure was not sensitive enough for our behavioral conditions and video recordings, we kept our manual, double blinded video analysis.

Figure 2d figure inset seems to be wrong, should the red line be the IgG group?

AR: Thanks to the referee to have pointed this mistake. It has been corrected in the new version of the MS.

The resolution of the figures is relatively low, especially in figure3.

AR: We do have better resolution figures as they were all based on vectorial drawing (Adobe illustrator). The poor quality should be improved when being allowed to download bigger size content, if the MS is accepted for publication.

Line 175 to 192, unless this is a new technique to the field, it seems more to belong to the method part, also I suggest figure4 a-c be moved to supplementary figure if it's not directly related to the result. This might be a personal preference, so feel free to ignore it.

AR: That was a matter of internal debate also. The recording of SPWR *in situ* is not as trivial as it sounds, as only been done in a few labs. It is also sometimes easy to mixed with other sources of baseline fluctuations, so we consider important to show at least once in the main figure the quality of the recordings. We therefore choose to maintain it in the main figure.

Figure 5b bottom, what is a,b,c?

AR: For clarity, we now indicated the time period that are concerned at the left side of the representative illustrations.

Figure 5c is a bit hard to follow, especially the bottom figure, what's the relationship between SWRs frequency and fEPSP amplitude? Although the author mentioned co-evolution, I wonder if it will make the readers confused about the relationship between SWR amplitude and fEPSP frequency as the percentage converged after 10 minutes, which as I understand, is just a coincidence?

AR: The convergence of potentiation to the same plateau (as expressed in %) is indeed a coincidence. We changed the sentence in the result part (page 8) for “Furthermore, we observed that the effect of HFS on synaptic strength and SPW-Rs frequency seems to have different time courses, the increase in evoked EPSP amplitude being detectable as early as in the 0-5 minutes post-tetanic period, whereas the effect on SPW-Rs frequency was not yet present (Figure 5b2 bottom, 5c). This possibly reflects an ongoing development of synaptic inputs potentiation, along with a progressive rise in CA3 cells excitability. Thus, these results suggested that the reinforcement of CA3→CA3 recurrent synapses increase CA3 region excitability and promotes the generation of ripples.”

Figure 6, I don't think it's a requested experiment to do, but I'm wondering if the authors have tried a similar experiment in CA1 and if it won't produce the same effect as in CA3.

AR: Indeed, we started the GluA2-AP KI strategy by targeting CA1 region. As can be seen below, 7 control and 5 KI-BIRA-NA animals were tested for DSA task. Coherent with CA3 data, control animals indeed exhibited less errors at sessions #5-6 than #1-2, a phenomenon that was not seen for the KI-BIRA-NA animals.

However, even if coordinates were significantly different: **CA1 experiments:** AP=-1,6/-2; ML=1,9/2,3; DV=-0,3 (+0,5 projection at injection time), **CA3 experiments:** AP=-2,15/-2,55; ML=2,45/2,85; DV=-1,4 (+0,5 projection at injection time), histological data were ambiguous in the case of CA1, as we did not successfully restricted the presence of NA in this region, and CA3 region often concerned by NA. Thus, because being ambiguous in their interpretation, we propose to keep these results as a referee figure (below). We would be ready to include it as a supplementary data if requested.

Reviewer Figure: An alternative AMPAR X-linking strategy targeting the dHPC CA1 area also blocked consolidation of DSA rule. **a:** We recently developed a new strategy for AMPAR X-linking. Knock-in mice expressing endogenous AP-tagged GluA2 AMPAR subunits can be biotinylated in presence of BirA^{ER}, and once exported to the cell surface can be immobilized in presence of external neutravidin (NA, cross-linking condition). **b:** *in vivo* pharmacological experiments were performed, combining early stereotaxic dHPC injections of AAV-BiRA-GFP or AAV-GFP, and pre-rest injections of saline or NA. **c:** histological controls for the GFP/BiRA GFP

fluorescence (Green) and the NA staining (red). Note that AMPAR immobilization is not restricted to the CA1 area. d: Mean error rates were compared between session#1-2 and session#5-6 to evaluate the retention of the DSA rule upon various pharmacological treatments (as indicated by colour coding). Paired t-test were used. ns: not significant, *: $p < 0.05$.

Sup Figure1:

Is red and blue indicate FaB and IgG here as well? Also, figure legend has two 'session's, which seems to be a typo.

AR: Thanks to the referee to have pointed this missing info. Indeed, the color code is the same as previous figures. It has been indicated in the new version of the MS.

Sup Figure2:

The label of sub-figure is misleading, where is c? And I think d is the subfigure c? and ii and iii are e and f?

AR: Thanks to the referee to have pointed this mistake. The legend was associated with another format of the figure. It has been corrected in the new version of the MS.

Reviewer #2 (Remarks to the Author):

This study presents data on how post-synaptic AMPAR mobility at CA3 recurrent synapses supporting the generation of ripples necessary for rule consolidation. When a rule has been encoded, a strong impact of AMPAR immobilization on ripples measured by in vivo recordings during resting periods. In situ examination of the interplay between AMPAR mobility, the results showed post-synaptic plasticity at CA3-CA3 recurrent synapses support ripple generation. The study makes an addition to a growing literature on hippocampal synaptic plasticity and memory consolidation.

Why did the authors investigate the role of AMPAR in synaptic plasticity of the memory consolidation, but not the NMDAR or any other receptors? The importance of AMPAR and the reason for this study are needed in the introduction.

AR: Indeed, this rational was missing.

The following paragraph has been added in the introduction (page 3): “20 years ago, we and others uncovered that AMPAR, along with NMDA and GABA_A receptors, are highly mobile at the neuronal surface, and are reversibly stabilized at synaptic site due to protein/protein interactions with various synaptic partners¹⁴. Recently, we showed that immobilization of GluA2-containing AMPAR leads to blockade of long-term potentiation¹⁵ without affecting basal synaptic transmission, offering a tool to assess the role of hippocampal AMPAR mobility (AMPARM) in the various phases of recent memories.”

Line 224, the authors made the conclusion “the effect of HFS on synaptic strength and SPW-Rs frequency seems to be temporally disconnected”, because the fEPSP and ripples were not enhanced simultaneously after the HFS. However, the mechanism of ripple generation is complicated, which is not only modulated by the presynaptic high frequency stimulation. So this conclusion should be toned down, because whether they are “temporally disconnected” cannot not only be measured by the increment in the first 5 mins. In addition, the unit was lack for “0-5 post-tetanic period” in Line 225.

AR: We modified the paragraph, and correct for the missing unit.

It is now (page 8): “Furthermore, we observed that the effect of HFS on synaptic strength and SPW-Rs frequency seems to have different time courses, the increase in evoked EPSP amplitude being detectable as early as in the 0-5 minutes post-tetanic period, whereas the effect on SPW-Rs frequency was not yet present (Figure 5b2 bottom, 5c). This possibly reflects an ongoing development of synaptic inputs potentiation, along with a progressive rise in CA3 cells excitability. Thus, these results suggested that the reinforcement of CA3→CA3 recurrent synapses increase CA3 region excitability and promotes the generation of ripples.”

Line 587, “Yellow zone: time after HFS application” cannot be found in the figure.

AR: Thanks to the referee to have pointed these mistakes. Indeed, in the Fig5 legend, some labels dedicated to an older version remained. This has been corrected in the new version of the manuscript.

In Fig1c, a subfigure was missing according to the figure legend.

AR: Thanks to the referee to have pointed these mistakes. The Fig1 legend has been change to: “Two cannulas were implanted above the dHPC, and anti GluA2-IgGs or control compounds were injected (top). In order to cover a large portion of the dHPC, multiple injection points were used in the ventro-dorsal axis. Entry points of the injection cannulas were mostly above the CA1 area (bottom).”

In Fig2, b-e showed the data repeatedly, which is not necessary.

AR: The purpose of figure 2 is to illustrate the similarity in the behavioral profiles of mice that discover the rule for the first time, and those of mice that are tested in session #5 after pre-learning injections of anti-GluA2 antibodies. In panel 2b, only averages of all mice are shown for session #1 and #5. As we know, an average can be similar for very diverging population distributions and thus it sounds important to shown in figure 5e that the distribution of IgG treated session #1 and #5 were strikingly similar, suggesting that animals returned to their initial “naïve” state according to the DSA rule.

In the Fig2b, the marker of VTE runs and no VTE runs look very similar.

AR: We have modified the symbols to allow better identification of the various groups in figure 2.

In Fig3a bottom, the y-label is not correct. It should be like ripple occurrence rate (Hz).

AR: Thanks to the referee to have pointed this ambiguity. It has been corrected in the new version of the MS.

In the analysis of Fig3b-right and Fig 3e, because the ripple amplitude has been inhibited by IgG, if the same criterial was used to detect ripples, the ripple frequency was certainly decreased. How to rule out the effect of amplitude on ripple detection?

AR: Thanks to the referee to have pointed this possibility. The detection and quantification of ripples is a tricky challenge, especially if comparing between different sessions. Indeed, even if recording conditions are preserved, including the noise level, the amount of sleep can be different and therefore salient events such as ripple, that occur during SWS, can differ between D-1 and D1, and thus contribute differentially to the threshold definition.

Actually, the threshold was recalculated independently on each day. The threshold for peak detection is the peak amplitude and peak timestamp for events differing for more than 5 SD from the mean of the absolute band-passed trace, thus there is not a fixed amplitude threshold at a given session. Thus indeed, if the amplitude of the salient events is decreasing strongly, it is possible that some events would go below the detection threshold. However, when considering the relationship between event amplitude and event frequency (see figure below), we observed that for the vast majority of the recordings, the amplitude of detected Ripples is not predictive for their frequency. Only for ripples below 0,1 mV in mean amplitude we would expect a strong decrease in frequency. However, in the 7 DSA+IgG animals, the ripple frequency was below than what would be expected for ripple of this mean amplitude (see the red line fit curve in the figure below). For us it suggests that if we cannot exclude a contribution of the decrease in amplitude to the decrease in frequency, it is not a major confounding factor to this particular index.

A sentence has been added in the discussion section (page 11): “Importantly, the relationship between ripple amplitude and frequency was modified in D1 in the DSA+IgG condition: indeed, ripple frequencies were lower than expected for events of these amplitudes (data not shown). This suggests that the amplitude change has only a minimal contribution to the frequency change.”

Figure legend: From in vivo recording data, we isolated the relationship existing between the amplitude and the frequency of recorded ripples, and showed separately those recorded before (D-1) and after (D1) injection of IgG and DSA learning (Data from figure 3 c-f). Interestingly, in D1, no clear relationship exists between the amplitude and the frequency, indicating that our selection procedure that is based on a threshold at 5SD above the mean is robust. Importantly, after DSA+IgG treatment, the curve established below the one of control conditions (grey line) and their D-1 controls (Orange line), supporting the fact that the low frequency cannot be attributed to event amplitude.

Also, the examples in Fig 3b-right was not consistent with this conclusion.

AR: We do not really understand where was the inconsistency. Graphs are showing the ripples frequency in D1 in various conditions. In the illustration for the DSA+IgG condition, no large increase in ripple frequency is present during slow wave sleep, which is quite different from all other tested conditions.

Line 227-228, again, this conclusion should be made more cautiously. The response of synaptic plasticity occurred earlier than that of the ripples, doesn't necessarily point to the causality between them.

AR: We modified the paragraph, and correct for the missing unit.

It is now (page 8): “Furthermore, we observed that the effect of HFS on synaptic strength and SPW-Rs frequency seems to have different time courses, the increase in evoked EPSP amplitude

being detectable as early as in the 0-5 minutes post-tetanic period, whereas the effect on SPW-Rs frequency was not yet present (Figure 5b2 bottom, 5c). This possibly reflects an ongoing development of synaptic inputs potentiation, along with a progressive rise in CA3 cells excitability. Thus, these results suggested that the reinforcement of CA3→CA3 recurrent synapses increase CA3 region excitability and promotes the generation of ripples.”

Line 267, the results showed that the accuracy of VTE runs was improved in control mice. However, the error rate was higher in VTE5-6 than in VTE 1-2 in Fig 6e. Is there a mistake on the labels in the figure?

AR: Indeed! Thanks to the referee to have pointed this exchange. It has been corrected in the new version of the MS.

In the Fig 8, the synaptic mechanism in the cortex was included in the model, however, there was no findings related to the cortex in this study.

AR: The referee is right in pointing this. This cortical part has been removed and we adapted the summary scheme that in now proposed as new figure 8.

There is no description for statistics. The stats (hypothesis tests) in all figures should be shown. For example, what's the stats in fig 1D? There is no main effect or interaction for 2-way ANOVA, but only the multiple comparison was labeled in the figure. What is the sample size N in Fig 1g? It the N too small to do stats?

AR: The referee is right in pointing this. We now provide this information in the figure legends, and summarize them in a table provided as a supplementary data (supplementary table) added to the MS.

REVIEWERS' COMMENTS

Reviewer #2 (Remarks to the Author):

The authors properly addressed my comments and have revised the manuscript accordingly. I think this manuscript is good for publication.